# Unleashing the Power of Graph Data Augmentation on Covariate Distribution Shift

**Yongduo Sui[1†], Qitian Wu[2], Jiancan Wu[1], Qing Cui[3], Longfei Li[3], Jun Zhou[3*],**
**Xiang Wang[1*], Xiangnan He[1*]**
[1]University of Science and Technology of China,
[2]Shanghai Jiao Tong University, [3]Ant Group
`syd2019@mail.ustc.edu.cn, echo740@sjtu.edu.cn,`
`{wujcan,xiangwang1223,xiangnanhe}@gmail.com,`
`{cuiqing.cq,longyao.llf,jun.zhoujun}@antgroup.com`

## Abstract

The issue of distribution shifts is emerging as a critical concern in graph representation learning. From the perspective of invariant learning and stable learning, a recently well-established paradigm for out-of-distribution generalization, stable features of the graph are assumed to causally determine labels, while environmental features tend to be unstable and can lead to the two primary types of distribution shifts. The correlation shift is often caused by the spurious correlation between environmental features and labels that differs between the training and test data; the covariate shift often stems from the presence of new environmental features in test data. However, most strategies, such as invariant learning or graph augmentation, typically struggle with limited training environments or perturbed stable features, thus exposing limitations in handling the problem of covariate shift. To address this challenge, we propose a simple-yet-effective data augmentation strategy, Adversarial Invariant Augmentation (AIA), to handle the covariate shift on graphs. Specifically, given the training data, AIA aims to extrapolate and generate new environments, while concurrently preserving the original stable features during the augmentation process. Such a design equips the graph classification model with an enhanced capability to identify stable features in new environments, thereby effectively tackling the covariate shift in data. Extensive experiments with in-depth empirical analysis demonstrate the superiority of our approach. The implementation codes are publicly available at `https://github.com/yongduosui/AIA`.

## 1 Introduction

While recent advances have made solid progress in learning effective representations for graph-structured data, most of the existing approaches operate under the assumption that training and test graphs are independently drawn from an identical distribution [1, 2]. However, this assumption often falls short in real-world scenarios due to the out-of-distribution (OOD) data that potentially exists during the testing phase [3], which results in distribution shifts between training and test graphs. As a result, there is increasing research interest in OOD generalization on graphs or learning with distribution shifts on graphs [4]. Some of the typical recent works attempt to build effective methods for handling general distribution shifts on graphs, from (causal) invariant learning [5, 3], model architecture designs [6], and data augmentation [7].

---

[†]This work was done during author's internship at Ant Group.
[*]Corresponding authors. Xiang Wang and Xiangnan He are also affiliated with Institute of Artificial Intelligence, Institute of Dataspace, Hefei Comprehensive National Science Center.

37th Conference on Neural Information Processing Systems (NeurIPS 2023).

With more specific views, other attempts focus on designing generalizable models for tackling distribution shifts in particular domains with certain data formats, *e.g.,* molecular graphs [8], recommender systems [9, 10], and anomaly detection [11].

However, the majority of existing studies primarily focus on the correlation shift, one type of distribution shift concerning OOD generalization [12, 13], leaving another equally important type of distribution shift, *i.e.,* the covariate shift, largely under-explored in graph representation learning. From the perspective of invariant learning and stable learning [14, 15, 16], covariate shift is in stark contrast to correlation shift *w.r.t.* stable and environmental features of graph data. Specifically, according to the commonly-used graph generation hypothesis in prior studies [17, 18, 5, 8], there often exist stable features, which are informative features of the entire graphs and can reflect the predictive patterns in data. Based on this, the relationship between stable

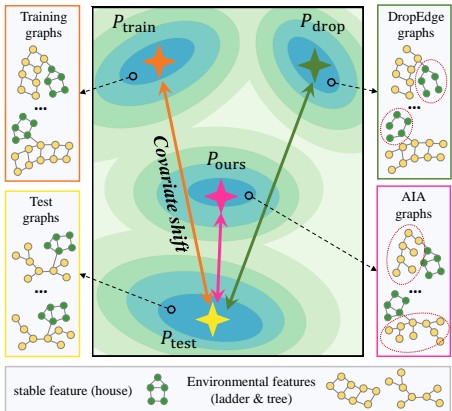

Figure 1: $P_{\text{train}}$ and $P_{\text{test}}$ denote the training and test distributions. $P_{\text{drop}}$ and $P_{\text{ours}}$ represent the distributions of augmented data via DropEdge and AIA.

features and labels is assumed to be invariant across environments. The remaining features could be unstable and varying across different environments, which mainly causes the following two distribution shifts: (1) correlation shift indicates that environments and labels establish inconsistent statistical correlations in training and test data, under the assumption that test environments are covered in the training data; whereas, (2) covariate shift characterizes that the environmental features in test data are unseen in training data [2, 12].

Considering a toy example in Figure 1, the environmental features *ladder* and *tree* are different in training and test data, which forms the covariate shift. Taking molecular property predictions as another example, functional groups (*e.g.,* nitrogen dioxide ($NO_2$)) are stable to determine the predictive property of molecules [5, 8]. Whereas, scaffolds (*e.g.,* carbon rings) are usually patterns irrelevant to the molecule properties, which can be seen as environmental features [2, 19]. In practice, we often need to use molecules collected in the past to train models, expecting that the models can predict the properties of molecules with new scaffolds in the future [20, 2, 19].

Considering the differences between correlation and covariate shifts, we take a close look into the existing efforts on graph generalization. They mainly fall into the following research lines, each of which has inherent limitations to solving the covariate shift. *i) Invariant Graph Learning* [17, 18, 5, 21] recently becomes a prevalent paradigm for OOD generalization. The basic idea is to capture stable features by minimizing the empirical risks in different environments. Unfortunately, it implicitly makes a prior assumption that test environments are available during training. This assumption is unrealistic owing to the obstacle of training data covering all possible test environments. Learning in limited environments can alleviate the spurious correlations that are hidden in the training data, but fail to extrapolate the test data with unseen environments. *ii) Graph Data Augmentation* [22, 23] perturbs graph features to enrich the distribution seen during training for better generalization. It can be roughly divided into node-level [24], edge-level [25], and graph-level [26, 7] with random [27] or adversarial strategies [28]. However, blindly augmenting the graphs can presumably destroys the stable features, and makes the augmented distributions out of control. For example, in Figure 1, the random strategy of DropEdge [25] will inevitably perturb the stable features (highlighted by red circles). As such, it may not sufficiently address the covariate shift and could potentially affect the generalization ability. Hence, we naturally ask a question: "*Compared to the training data, can we generate new data that satisfy two conditions: 1) having new environments; 2) keeping the original stable features unchanged?*"

Towards this end, we introduce two intuitive principles for graph augmentation: environmental feature discrepancy and stable feature consistency. The discrepancy principle promotes the exploration of new environments beyond the scope of training data, while the consistency principle seeks to maintain the integrity of stable features during augmentation. In order to achieve these principles, we devise a simple yet effective graph augmentation strategy: Adversarial Invariant Augmentation (AIA). Specifically, we employ an adversarial augmenter, a network that augments graphs by adversarially generating masks on them, thereby facilitating OOD exploration to enhance environmental discrep-

ancy. To foster stable feature consistency, we use another network, *i.e.,* stable feature generator, which constructs masks that encapsulate stable features. We then delicately merge these masks and apply them to the graph data. As depicted in Figure 1, AIA primarily augments environmental features while leaving the stable elements unchanged. Our approach equips the graph classifier model with an enhanced ability to identify stable features in new environments and effectively mitigate the covariate shift issue. We also conduct extensive experiments and in-depth analyses. The experimental results highlight the limitations of several previous methods and underscore the superiority of our method in addressing the covariate shift issue, providing empirical support for our claims.

## 2   Preliminaries

We define the uppercase letters (*e.g.,* $G$) as random variables. The lower-case letters (*e.g.,* $g$) are samples of variables, and the blackboard bold typefaces (*e.g.,* $\mathbb{G}$) denote the sample spaces. Let $g = (\mathbf{A}, \mathbf{X}) \in \mathbb{G}$ denote a graph, where $\mathbf{A}$ and $\mathbf{X}$ are its adjacency matrix and node features, respectively. It is assigned with a label $y \in \mathbb{Y}$ with a fixed labeling rule $\mathbb{G} \to \mathbb{Y}$. Let $\mathcal{D} = \{(g_i, y_i)\}$ denote a dataset that is divided into a training set $\mathcal{D}_{\mathrm{tr}} = \{(g_i^e, y_i^e)\}_{e \in \mathcal{E}_{\mathrm{tr}}}$ and a test set $\mathcal{D}_{\mathrm{te}} = \{(g_i^e, y_i^e)\}_{e \in \mathcal{E}_{\mathrm{te}}}$. $\mathcal{E}_{\mathrm{tr}}$ and $\mathcal{E}_{\mathrm{te}}$ are the index sets of training and test environments, respectively.

### 2.1   Definitions and Problem Formations

In this work, we focus on the graph classification scenario, which aims to train models with $\mathcal{D}_{\mathrm{tr}}$ and infer the labels in $\mathcal{D}_{\mathrm{te}}$, such as molecular property prediction. From the viewpoints of invariant learning and stable learning, the inner mechanism of the labeling rule $\mathbb{G} \to \mathbb{Y}$ is usually assumed to depend on the stable features [17, 18, 5, 8, 21], which are informative substructures of the entire graph. The relationship between the stable features and labels is assumed to be invariant across different environments, which makes OOD generalization possible [12]. Environmental features in graph data are assumed to have no causal-effect on labels. For instance, the chemical properties of molecules are mainly determined by specific functional groups, which can be regarded as stable features [17, 5, 19, 8]. Conversely, their scaffold structures, often irrelevant to their properties, can be seen as environmental features [20, 8].

Due to the instability of environmental features and the limitations in the data collection process, the training and test distributions are often inconsistent in real-world scenarios, *i.e.,* $P_{\mathrm{tr}}(G, Y) \neq P_{\mathrm{te}}(G, Y)$, which leads to two main types of distribution shifts [2, 12]: (1) Correlation shift (*aka.* spurious correlation or concept shift [2]) refers to $P_{\mathrm{tr}}(G|Y) \neq P_{\mathrm{te}}(G|Y), P_{\mathrm{tr}}(G) = P_{\mathrm{te}}(G)$. It indicates that there exist spurious statistical correlations in the training data, while these correlations might not hold in the test data. (2) Covariate shift denotes $P_{\mathrm{tr}}(G|Y) = P_{\mathrm{te}}(G|Y), P_{\mathrm{tr}}(G) \neq P_{\mathrm{te}}(G)$, which means that there exist new features, *e.g.,* environmental features, in the test data. It may be ascribed to either insufficient quantity or diversity of data in the training set, as well as the unknown characteristics of the test environments. We provide formal definitions and examples of these two distribution shifts in Appendix A. Here, inspired by the prior study [12], we offer a formal definition to measure the graph covariate shift.

**Definition 2.1 (Graph Covariate Shift)** *Let $P_{\mathrm{tr}}$ and $P_{\mathrm{te}}$ denote the probability functions of the training and test distributions. We measure the covariate shift between distributions $P_{\mathrm{tr}}$ and $P_{\mathrm{te}}$ as*

$$\mathrm{GCS}(P_{\mathrm{tr}}, P_{\mathrm{te}}) = \frac{1}{2} \int_{\mathcal{S}} |P_{\mathrm{tr}}(g) - P_{\mathrm{te}}(g)| dg, \tag{1}$$

*where $\mathcal{S} = \{g \in \mathbb{G} \mid P_{\mathrm{tr}}(g) \cdot P_{\mathrm{te}}(g) = 0\}$, which covers the features (e.g., environmental features) that do not overlap between the two distributions.*

$\mathrm{GCS}(\cdot, \cdot)$ is always bounded in $[0, 1]$. The issue of graph covariate shift is very common in practice. For example, we often need to train models on past molecular graphs, and hope that the model can predict the chemical properties of future molecules with new features, *e.g.,* new scaffolds [20]. In addressing the graph OOD issue, a majority of the strategies [17, 5, 18, 29, 21] grounded in invariant graph learning aim to pinpoint stable or invariant features. This is mainly accomplished by minimizing empirical risks across an array of training environments. Nonetheless, these methodologies frequently operate under the assumption of a shared input space across training and test data. This assumption, however, often fails on covariate shift. It presents substantial challenges for these models when it

comes to accurately identifying stable features within new testing environments. Consequently, while these methods typically exhibit satisfactory performance in managing correlation shifts, they often underperform in the face of covariate shifts. In this work, we focus on the covariate shift issue in graph classification, and we also give a formal definition of this problem as follows.

**Problem 2.2 (Graph Classification under Covariate Shift)** *Given the training and test sets with environment sets $\mathcal{E}_{\text{tr}}$ and $\mathcal{E}_{\text{te}}$, they follow distributions $P_{\text{tr}}$ and $P_{\text{te}}$, and satisfy:* $\text{GCS}(P_{\text{tr}}, P_{\text{te}}) > \epsilon$, *where $\epsilon \in (0, 1)$ represents the degree of covariate shift. We aim to use the data collected from training environments $\mathcal{E}_{\text{tr}}$, and learn a powerful graph classifier $f^* : \mathbb{G} \to \mathbb{Y}$ that performs well in all possible test environments $\mathcal{E}_{\text{te}}$:*

$$f^* = \arg\min_f \ \sup_{e \in \mathcal{E}_{\text{te}}} \mathbb{E}^e[\ell(f(g), y)], \tag{2}$$

*where $\mathbb{E}^e[\ell(f(g), y)]$ is the empirical risk on the environment $e$, and $\ell(\cdot, \cdot)$ is the loss function.*

# 3 Methodology

To solve Problem 2.2, our idea is to generate new graphs through data augmentation. In this section, we first propose two principles for graph augmentation. Guided by these principles, we design a novel graph augmentation method, AIA, which can effectively address the covariate shift issue.

## 3.1 Two Principles for Graph Augmentation

We can observe that covariate shift is mainly caused by the scarcity of training environments. Hence, we first propose the discrepancy principle for graph augmentation.

**Principle 3.1 (Environmental Feature Discrepancy)** *Given a graph set $\{g\}$ with distribution function $P$, let $T(\cdot)$ denote an augmentation function that augments graphs $\{T(g)\}$ to distribution $\widetilde{P}$. Then $T(\cdot)$ should meet $\text{GCS}(P, \widetilde{P}) \to 1$.*

From the perspective of data distribution, it requires that $\widetilde{P}$ should keep away from the original distribution $P$. From the perspective of data instances, it emphasizes the discrepancy in the environments between the generated graphs and the original graphs. Since it does not give constraints on stable features, we here propose the second principle for graph augmentation.

**Principle 3.2 (Stable Feature Consistency)** *Given a set of graphs $\{g\}$ with a corresponding stable feature set $\{g_{\text{sta}} = (\mathbf{A}_{\text{sta}}, \mathbf{X}_{\text{sta}})\}$. Let $T(\cdot)$ denote an augmentation function that augments graphs $\{T(g)\}$ with a corresponding stable feature set $\{\widetilde{g}_{\text{sta}} = (\widetilde{\mathbf{A}}_{\text{sta}}, \widetilde{\mathbf{X}}_{\text{sta}})\}$. Then $T(\cdot)$ should meet $\mathbb{E}[\|\mathbf{A}_{\text{sta}} - \widetilde{\mathbf{A}}_{\text{sta}}\|_F^2] \to 0$ and $\mathbb{E}[\|\mathbf{X}_{\text{sta}} - \widetilde{\mathbf{X}}_{\text{sta}}\|_F^2] \to 0$, where $\|\cdot\|_F$ is the Frobenius norm.*

It necessitates that the stable features of the generated graphs should maintain consistency with those of the original graphs. This principle ensures the preservation of these stable features within the original training data, thereby safeguarding sufficient information pertaining to the labels. Consequently, this principle enhances the potential for generalization.

## 3.2 Out-of-distribution Exploration

Given a GNN model $f(\cdot)$ with parameters $\theta$, we decompose $f = \Phi \circ h$, where $h(\cdot) : \mathbb{G} \to \mathbb{R}^d$ is a graph encoder to yield $d$-dimensional representations, and $\Phi(\cdot) : \mathbb{R}^d \to \mathbb{Y}$ is a classifier. To comply with Principle 3.1, we need to do OOD exploration. Inspired by distributionally robust optimization [30, 31], we consider the following optimization objective:

$$\min_\theta \left\{ \sup_{\widetilde{P}} \{ \mathbb{E}_{\widetilde{P}}[\ell(f(g), y)] : D(\widetilde{P}, P) \le \rho \} \right\}, \tag{3}$$

where $P$ and $\widetilde{P}$ are the original and explored data distributions, respectively. $D(\cdot, \cdot)$ is a distance metric between two probability distributions. The solution to Equation (3) guarantees the generalization within a robust radius $\rho$ of the distribution $P$. To better measure the distance between distributions,

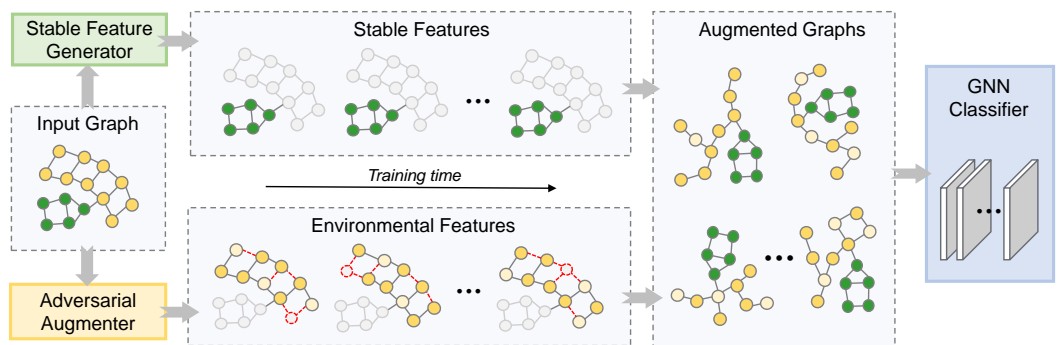

Figure 2: The overview of Adversarial Invariant Augmentation (AIA) Framework.

as suggested by [32], we adopt the Wasserstein distance [33, 34] as the distance metric. The distance metric function can be defined as $D(\widetilde{P}, P) = \inf_{\mu \in \Gamma(\widetilde{P}, P)} \mathbb{E}_\mu[c(\widetilde{g}, g)]$, where $\Gamma(\widetilde{P}, P)$ is the set of all couplings of $\widetilde{P}$ and $P$; $c(\cdot, \cdot)$ is the cost function. Studies [35, 34] also suggest that the distances in representation space typically correspond to semantic distances. Hence, we define the cost function in the representation space and give the transportation cost as $c(\widetilde{g}, g) = \|h(\widetilde{g}) - h(g)\|_2^2$. It denotes the "cost" of augmenting the graph $g$ to $\widetilde{g}$. We can observe that it is difficult to set a proper $\rho$. Instead, we consider the Lagrangian relaxation for a fixed penalty coefficient $\gamma$. Inspired by [32], we can reformulate Equation (3) as follows:

$$\min_\theta \left\{ \sup_{\widetilde{P}} \{ \mathbb{E}_{\widetilde{P}}[\ell(f(g), y)] - \gamma D(\widetilde{P}, P) \} = \mathbb{E}_P[\phi(f(g), y)] \right\}, \tag{4}$$

where $\phi(f(g), y) := \sup_{\widetilde{g} \in \mathbb{G}} \{ \ell(f(\widetilde{g}), y) - \gamma c(\widetilde{g}, g) \}$. And we define $\phi(f(g), y)$ as the robust surrogate loss. If we conduct gradient descent on the robust surrogate loss, we will have:

$$\nabla_\theta \phi(f(g), y) = \nabla_\theta \ell(f(\widetilde{g}^*), y), \quad \text{where} \quad \widetilde{g}^* = \arg\max_{\widetilde{g} \in \mathbb{G}} \{ \ell(f(\widetilde{g}), y) - \gamma c(\widetilde{g}, g) \}. \tag{5}$$

$\widetilde{g}^*$ is an augmented view of the original data $g$. Hence, to achieve OOD exploration, we need to perform graph data augmentation via Equation (5) on the original data $g$.

### 3.3 Implementations of AIA

Equation (5) endows the ability of OOD exploration to data augmentation, which makes the augmented data meet the discrepancy principle. To achieve the consistency principle, we also need to capture stable features. Hence, we design a graph augmentation strategy: Adversarial Invariant Augmentation (AIA). The overview of the proposed framework is depicted in Figure 2, which mainly consists of two components: adversarial augmenter and stable feature generator. Adversarial augmenter achieves OOD exploration through adversarial data augmentation; meanwhile, the stable feature generator keeps stable feature consistency by identifying stable features from data. Below we elaborate on the implementation details.

**Adversarial Augmenter & Stable Feature Generator.** We design two networks, adversarial augmenter $T_{\theta_1}(\cdot)$ and stable feature generator $T_{\theta_2}(\cdot)$, which generate masks for nodes and edges of graphs. They have the same structure and are parameterized by $\theta_1$ and $\theta_2$, respectively. Given an input graph $g = (\mathbf{A}, \mathbf{X})$ with $n$ nodes, mask generation network first obtains the node representations via a GNN encoder $\widetilde{h}(\cdot)$. To judge the importance of nodes and edges, it adopts two MLP networks $\mathrm{MLP}_1(\cdot)$ and $\mathrm{MLP}_2(\cdot)$ to generate the soft node mask matrix $\mathbf{M}^x \in \mathbb{R}^{n \times 1}$ and edge mask matrix $\mathbf{M}^a \in \mathbb{R}^{n \times n}$ for graph data, respectively. In summary, the mask generation network can be decomposed as:

$$\mathbf{Z} = \widetilde{h}(g), \quad \mathbf{M}_i^x = \sigma(\mathrm{MLP}_1(\mathbf{h}_i)), \quad \mathbf{M}_{ij}^a = \sigma(\mathrm{MLP}_2([\mathbf{z}_i, \mathbf{z}_j])), \tag{6}$$

where $\mathbf{Z} \in \mathbb{R}^{n \times d}$ is node representation matrix, whose $i$-th row $\mathbf{z}_i = \mathbf{Z}[i, :]$ denotes the representation of node $i$, and $\sigma(\cdot)$ is the sigmoid function that maps the mask values $\mathbf{M}_i^x$ and $\mathbf{M}_{ij}^a$ to $[0, 1]$.

**Adversarial Invariant Augmentation.** To estimate $\widetilde{g}^*$ in Equation (5), we define the following adversarial learning objective:

$$\max_{\theta_1} \{ \mathcal{L}_{\mathrm{adv}} = \mathbb{E}_{P_{\mathrm{tr}}}[\ell(f(T_{\theta_1}(g)), y) - \gamma c(T_{\theta_1}(g), g)] \}. \tag{7}$$

Then we can augments the graph by $T_{\theta_1}(g) = (\mathbf{A} \odot \mathbf{M}_{\mathrm{adv}}^a, \mathbf{X} \odot \mathbf{M}_{\mathrm{adv}}^x)$, where $\odot$ is the broadcasted element-wise product. Although adversarially augmented graphs guarantee environmental discrepancy, it might destroys the stable parts. Therefore, we utilize the stable feature generator $T_{\theta_2}(\cdot)$ to capture stable features and combine them with different environmental features. Following the sufficiency and invariance conditions [17, 3, 18, 5, 8, 29], we define the stable feature learning objective as:

$$\min_{\theta, \theta_2} \{ \mathcal{L}_{\mathrm{sta}} = \mathbb{E}_{P_{\mathrm{tr}}}[\ell(f(T_{\theta_2}(g)), y) + \ell(f(\widetilde{g}), y)] \}, \tag{8}$$

where $\widetilde{g} = (\mathbf{A} \odot \widetilde{\mathbf{M}}^a, \mathbf{X} \odot \widetilde{\mathbf{M}}^x)$ is the augmented graph. It adopts the mask combination strategy: $\widetilde{\mathbf{M}}^a = (\mathbf{1}^a - \mathbf{M}_{\mathrm{sta}}^a) \odot \mathbf{M}_{\mathrm{adv}}^a + \mathbf{M}_{\mathrm{sta}}^a$ and $\widetilde{\mathbf{M}}^x = (\mathbf{1}^x - \mathbf{M}_{\mathrm{sta}}^x) \odot \mathbf{M}_{\mathrm{adv}}^x + \mathbf{M}_{\mathrm{sta}}^x$, where $\mathbf{M}_{\mathrm{sta}}^a$ and $\mathbf{M}_{\mathrm{sta}}^x$ are generated by $T_{\theta_2}(\cdot)$, $\mathbf{1}^a$ and $\mathbf{1}^x$ are all-one matrices, and if there is no edge between node $i$ and node $j$, then we set $\mathbf{1}_{ij}^a$ to 0. Now we explain this combination strategy. Taking $\widetilde{\mathbf{M}}^x$ as an example, since $\mathbf{M}_{\mathrm{sta}}^x$ denotes the captured stable regions via $T_{\theta_2}(\cdot)$, $\mathbf{1}^x - \mathbf{M}_{\mathrm{sta}}^x$ represents the complementary parts, which are environmental regions. $\mathbf{M}_{\mathrm{adv}}^x$ represents the adversarial perturbation, so $(\mathbf{1}^x - \mathbf{M}_{\mathrm{sta}}^x) \odot \mathbf{M}_{\mathrm{adv}}^x$ is equivalent to applying the adversarial perturbation on environmental features, meanwhile, sheltering the stable features. Finally, $+\mathbf{M}_{\mathrm{sta}}$ signifies that the augmented data should preserve the original stable features. Consequently, it satisfies both principles. Upon analysis of Equation (8), the first term implies that the stable features are sufficient for making right predictions. The second term promotes right and invariant predictions under generated environments utilizing stable features.

**Regularization.** For Equation (7), the adversarial optimization tends to remove more nodes and edges, so we should also constrain the perturbations. Although Equation (8) satisfies the sufficiency and invariance conditions, it is necessary to impose constraints on the ratio of the stable features to prevent trivial solutions. Hence, we first define the regularization function $r(\mathbf{M}, k, \lambda) = (\sum_{ij} \mathbf{M}_{ij}/k - \lambda) + (\sum_{ij} \mathbb{I}[\mathbf{M}_{ij} > 0]/k - \lambda)$, where $k$ is the total number of elements to be constrained, $\mathbb{I} \in \{0, 1\}$ is an indicator function. The first term penalizes the average ratio close to $\lambda$, while the second term encourages an uneven distribution. Given a graph with $n$ nodes and $m$ edges, we define the regularization term for adversarial augmentation and stable feature learning as:

$$\mathcal{L}_{\mathrm{reg}_1} = \mathbb{E}_{P_{\mathrm{tr}}}[r(\mathbf{M}_{\mathrm{adv}}^x, n, \lambda_a) + r(\mathbf{M}_{\mathrm{adv}}^a, m, \lambda_a)], \tag{9}$$

$$\mathcal{L}_{\mathrm{reg}_2} = \mathbb{E}_{P_{\mathrm{tr}}}[r(\mathbf{M}_{\mathrm{sta}}^x, n, \lambda_s) + r(\mathbf{M}_{\mathrm{sta}}^a, m, \lambda_s)], \tag{10}$$

where $\lambda_s \in (0, 1)$ is the ratio of stable features, we usually set $\lambda_a = 1$ for adversarial learning, which can alleviate excessive perturbations. The algorithm is provided in Appendix D.1.

## 4   Theoretical Discussions

In this section, we engage in theoretical discussions to elucidate our learning objective and its connections with the covariate shift. We first explore the relationship between our optimization objective and the discrepancy principle. Recalling the optimization objective of Equation 3, we aspire to identify a distribution $\widetilde{P}$ that can manifest within a Wasserstein ball [36], which is centered on distribution $P$, with distance $\rho$ serving as the radius. Under appropriate conditions, we find that our learning optimization objective can establish a close connection with OOD exploration.

**Proposition 4.1** *Consider a probability distribution $P$ defined over a measurable space $(\Omega, \mathcal{F})$, where $\Omega$ denotes the sample space and $\mathcal{F}$ is a $\sigma$-algebra on $\Omega$. We construct two Wasserstein balls with $P$ at their center and radii $\rho_1$ and $\rho_2$ respectively. Utilizing Equation 3, we generate two distinct distributions, $\widetilde{P}_1$ and $\widetilde{P}_2$, within the space $(\Omega, \mathcal{F})$. If (i) $P$ is an isotropic distribution; (ii) $\exists \boldsymbol{x}_1, \boldsymbol{x}_2 \in \Omega$ such that $P(\boldsymbol{x}) = \widetilde{P}_1(\boldsymbol{x} - \boldsymbol{x}_1) = \widetilde{P}_2(\boldsymbol{x} - \boldsymbol{x}_2)$; (iii) $\rho_1 < \rho_2$, then we have $\mathrm{GCS}(P, \widetilde{P}_1) \leq \mathrm{GCS}(P, \widetilde{P}_2)$.*

This suggests that by appropriately increasing the robustness radius in AIA, we can effectively amplify the covariate shift between the training and generated distributions. This in turn underscores the reliability of our discrepancy principle, to a certain degree. Comprehensive proofs and detailed discussions supporting these conclusions can be found in Appendix B.

Table 1: Performance on synthetic and real-world datasets. Numbers in **bold** indicate the best performance, while the underlined numbers indicate the second best performance.

| Type | Method | Motif | | CMNIST | Molbbbp | | Molhiv | |
|---|---|---|---|---|---|---|---|---|
| | | base | size | color | scaffold | size | scaffold | size |
| General Generalization | ERM | 68.66±4.25 | 51.74±2.88 | 28.60±1.87 | 68.10±1.68 | 78.29±3.76 | 69.58±2.51 | 59.94±2.37 |
| | IRM | 70.65±4.17 | 51.41±3.78 | 27.83±2.13 | 67.22±1.15 | 77.56±2.48 | 67.97±1.84 | 59.00±2.92 |
| | GroupDRO | 68.24±8.92 | 51.95±5.86 | 29.07±3.14 | 66.47±2.39 | 79.27±2.43 | 70.64±2.57 | 58.98±2.16 |
| | VREx | 71.47±6.69 | 52.67±5.54 | 28.48±2.87 | 68.74±1.03 | 78.76±2.37 | 70.77±2.84 | 58.53±2.88 |
| Graph Generalization | DIR | 62.07±8.75 | 52.27±4.56 | 33.20±6.17 | 66.86±2.25 | 76.40±4.43 | 68.07±2.29 | 58.08±2.31 |
| | CAL | 65.63±4.29 | 51.18±5.60 | 27.99±3.24 | 68.06±2.60 | 79.50±4.81 | 67.37±3.61 | 57.95±2.24 |
| | GSAT | 62.80±11.41 | 53.20±8.35 | 28.17±1.26 | 66.78±1.45 | 75.63±3.83 | 68.66±1.35 | 58.06±1.98 |
| | OOD-GNN | 61.10±7.87 | 52.61±4.67 | 26.49±2.94 | 66.72±1.23 | 79.48±4.19 | 70.46±1.97 | 60.60±3.77 |
| | StableGNN | 57.07±14.10 | 46.93±8.85 | 28.38±3.49 | 66.74±1.30 | 77.47±4.69 | 68.44±1.33 | 56.71±2.79 |
| | CIGA | 66.43±11.31 | 49.14±8.34 | 32.22±2.67 | 64.92±2.09 | 65.98±3.31 | 69.40±2.39 | 59.55±2.56 |
| | DisC | 51.08±3.08 | 50.39±1.15 | 24.99±1.78 | 67.12±2.11 | 56.59±10.09 | 68.07±1.75 | 58.76±0.91 |
| Graph Augmentation | DropEdge | 45.08±4.46 | 45.63±4.61 | 22.65±2.90 | 66.49±1.55 | 78.32±3.44 | 70.78±1.38 | 58.53±1.26 |
| | GREA | 56.74±9.23 | 54.13±10.02 | 29.02±3.26 | 69.72±1.66 | 77.34±3.52 | 67.79±2.56 | 60.71±2.20 |
| | FLAG | 61.12±5.39 | 51.66±4.14 | 32.30±2.69 | 67.69±2.36 | 79.26±2.26 | 68.45±2.30 | 60.59±2.95 |
| | M-Mixup | 70.08±3.82 | 51.48±4.91 | 26.47±3.45 | 68.75±0.34 | 78.92±2.43 | 68.88±2.63 | 59.03±3.11 |
| | $\mathcal{G}$-Mixup | 59.66±7.03 | 52.81±6.73 | 31.85±5.82 | 67.44±1.62 | 78.55±4.16 | 70.01±2.52 | 59.34±2.43 |
| | AIA (ours) | **73.64±5.15** | **55.85±7.98** | **36.37±4.44** | **70.79±1.53** | **81.03±5.15** | **71.15±1.81** | **61.64±3.37** |

# 5 Experiments

In this section, we conduct extensive experiments to answer the following **R**esearch **Q**uestions:

- **RQ1:** Compared to existing efforts, how does AIA perform under covariate shift?
- **RQ2:** Can the proposed AIA achieve the principles of environmental feature discrepancy and stable feature consistency, thereby alleviating the graph covariate shift?
- **RQ3:** How do the different components and hyperparameters of AIA affect the performance?

## 5.1 Experimental Settings

**Datasets.** We use graph OOD datasets [2] and OGB datasets [20], which include Motif, CMNIST, Molbbbp, and Molhiv. Following [2], we adopt the base, color, size, and scaffold data splitting to create various covariate shifts. The details of the datasets, metrics, implementations, and other settings are provided in Appendix D.2 and D.4. More experiments are provided in Appendix E.

**Baselines.** We adopt 16 baselines, which can be divided into the following three specific categories:

- **General Generalization Algorithms:** ERM, IRM [14], GroupDRO [31], VREx [37].
- **Graph Generalization Algorithms:** DIR [17], CAL [5], GSAT [38], OOD-GNN [39], StableGNN [40], CIGA [41], DisC [21].
- **Graph Augmentation:** DropEdge [25], GREA [18], FLAG [24], M-Mixup [26], $\mathcal{G}$-Mixup [7].

## 5.2 Main Results (RQ1)

We first make comparisons with various baselines in Table 1, and have the following observations:

Most generalization and augmentation methods fail under covariate shift. VREx achieves a 2.81% improvement on Motif (base). For two shifts of Molhiv, data augmentation methods GREA and DropEdge obtain 1.20% and 0.77% improvements. The invariant learning methods, *i.e.,* DIR and CAL also obtain 4.60% and 1.53% improvements on CMNIST and Molbbbp (size). Unfortunately, none of the methods consistently outperform ERM. For example, GREA and DropEdge perform poorly on Motif (base), ↓11.92% and ↓23.58%. DIR and CAL also fail on Molhiv. These show that both invariant learning and data augmentation methods have their own weaknesses, which lead to unstable performance when facing complex and diverse covariate shifts from different datasets.

AIA consistently outperforms most baseline methods. Compared with ERM, AIA can obtain significant improvements. For two types of covariate shifts on Motif, AIA surpasses ERM by 4.98% and 4.11%, respectively. In contrast to the large performance variances on different datasets achieved by

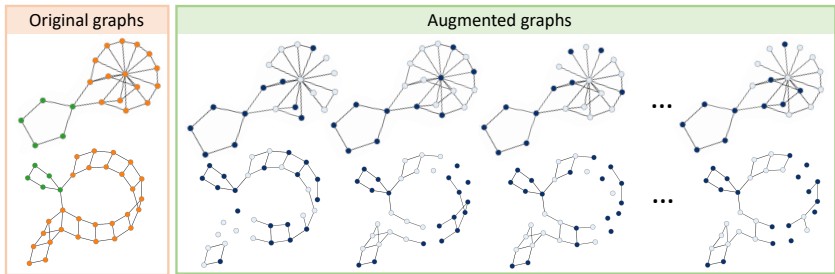

Figure 3: Visualizations of the augmented graphs via AIA.

Table 2: Covariate shift comparisons with different augmentation strategies.

| Method | Motif (base) | | Motif (size) | | CMNIST (color) | | Molbbbp (scaffold) | |
|---|---|---|---|---|---|---|---|---|
| | Aug-Train | Aug-Test | Aug-Train | Aug-Test | Aug-Train | Aug-Test | Aug-Train | Aug-Test |
| Original | 0 | $0.557_{\pm0.141}$ | 0 | $0.522_{\pm0.421}$ | 0 | $0.490_{\pm0.226}$ | 0 | $0.419_{\pm0.079}$ |
| DropEdge | $0.772_{\pm0.213}$ | $0.515_{\pm0.033}$ | $0.851_{\pm0.138}$ | $0.161_{\pm0.271}$ | $0.627_{\pm0.186}$ | $0.539_{\pm0.260}$ | $0.758_{\pm0.192}$ | $0.737_{\pm0.211}$ |
| FLAG | $0.001_{\pm0.001}$ | $0.533_{\pm0.016}$ | $0.002_{\pm0.018}$ | $0.507_{\pm0.121}$ | $0.003_{\pm0.002}$ | $0.442_{\pm0.062}$ | $0.001_{\pm0.001}$ | $0.413_{\pm0.088}$ |
| $\mathcal{G}$-Mixup | $0.690_{\pm0.186}$ | $0.472_{\pm0.043}$ | $0.816_{\pm0.154}$ | $0.299_{\pm0.343}$ | $0.408_{\pm0.228}$ | $0.351_{\pm0.318}$ | $0.551_{\pm0.258}$ | $0.545_{\pm0.231}$ |
| AIA (ours) | $0.369_{\pm0.169}$ | $0.462_{\pm0.063}$ | $0.649_{\pm0.143}$ | $0.098_{\pm0.070}$ | $0.516_{\pm0.106}$ | $0.307_{\pm0.108}$ | $0.422_{\pm0.049}$ | $0.393_{\pm0.028}$ |

baselines, AIA consistently obtains the leading performance across the board. For CMNIST, AIA achieves a performance improvement of 3.17% compared to the best baseline DIR. For Motif, the performance is improved by 2.17% and 1.72% compared to VREx and GREA. These results illustrate that AIA can overcome the shortcomings of invariant learning and data augmentation. Armed with the principles of environmental feature diversity and stable feature invariance, AIA achieves stable and consistent improvements on different datasets with various covariate shifts. In addition, although we focus on covariate shift in this work, we also carefully check the performance of AIA under correlation shift, and the results are presented in Appendix E.

### 5.3 In-depth Analyses (RQ2)

In this section, we conduct qualitative and quantitative experiments to support our two principles. Firstly, we utilize $GCS(\cdot, \cdot)$ as the measurement to quantify the degree of covariate shift. The detailed estimation procedure is provided in Appendix C. We select four different domains, *i.e.,* base, size, color and scaffold, to create covariate shifts. The experimental results are shown in Table 2. We calculated covariate shifts between the augmentation distribution $P_{aug}$ with the training $P_{tr}$ or test distribution $P_{te}$. "Aug-Train" and "Aug-Test" represent $GCS(P_{aug}, P_{tr})$ and $GCS(P_{aug}, P_{te})$, respectively. From the results in Table 2, we make these observations.

**Discrepancy Principle.** The term "Original" denotes the training distribution prior to augmentation. It's observed that substantial covariate shifts exist between the training and test distributions, ranging from 0.419 to 0.557. The DropEdge technique notably expands *Aug-Train*, with a range of 0.627 to 0.851, while concurrently increasing *Aug-Test*, as evidenced by CMNIST (0.490 to 0.539) and Molbbbp (0.419 to 0.737). However, a distribution that deviates excessively from the test distribution may not effectively address the issue of covariate shift. FLAG, which perturbs only the node features, yields minor values in both *Aug-Train* and *Aug-Test*. $\mathcal{G}$-Mixup notably augments *Aug-Train* by generating OOD samples, but doesn't necessarily limit *Aug-Test*. Finally, AIA extends the disparity with the training distribution by augmenting environmental features, signifying that AIA can adeptly implement the principle of environmental feature discrepancy. Simultaneously, the imposed consistency constraint on stable features restricts the generated distribution from straying too far from the test distribution, as observed in Motif-base (0.557 to 0.462), Motif-size (0.522 to 0.098), and CMNIST (0.490 to 0.307).

**Augmentation Diversity.** We further delve into the diversity of data augmentation. The concept of augmentation diversity stems from the intuition that augmentations with greater degrees of freedom yield better performance [42]. Accordingly, we propose conditional entropy to

Table 3: Augmentation Diversity.

| Method | Full Graph | Env. Feature | Sta. Feature |
|---|---|---|---|
| DropEdge | $0.999_{\pm0.065}$ | $0.933_{\pm0.029}$ | $0.971_{\pm0.067}$ |
| AIA (ours) | $0.561_{\pm0.223}$ | $0.508_{\pm0.136}$ | $0.259_{\pm0.106}$ |

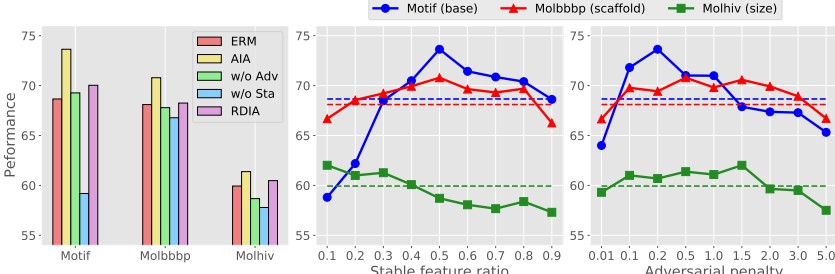

Figure 4: (*Left*): Different components in AIA. (*Middle*): Different ratios $\lambda_s$ of stable features. (*Right*): Different penalties $\gamma$. Dashed lines denote the ERM.

measure the diversity of generated data: $H(\tilde{G}|G) = -\mathbb{E}_G[\sum_{\tilde{g}} p(\tilde{g}|G)\log(p(\tilde{g}|G))]$, where $\tilde{G}$ and $G$ represent the generated graph and original graph, respectively. To substantiate our ability to manage perturbed environmental features while preserving stable features, we examine diversity at the feature level, *i.e.,* stable and environmental features. We employ the Motif dataset for validation due to its inclusion of ground-truth labels for stable and environmental features. The results in Table 3 reveal that our approach can guarantee the diversity of environmental features while constraining the variation of stable features.

To substantiate two principles of AIA, we present a selection of augmented graphs in Figure 3. These graphs are randomly sampled during the training phase. In the original Motif, the stable feature, is represented by the green portion and its type determines the label, while the yellow portion signifies the base-graph, or the environmental feature. Figure 3 (*Right*) exhibits the augmented samples generated during training. Nodes depicted in darker colors and edges with broader lines indicate higher soft-mask values. These results lead us to several noteworthy observations.

**Visualization Analyses.** AIA primarily perturbs the environmental features, while leaving the stable components undisturbed. In the Motif dataset, the base-graph represents a *ladder* and the motif-graph signifies a *house*. Following augmentation, the nodes and edges of the *ladder* graph undergo perturbations. However, the *house* component remains consistently stable throughout the training process. It shows that AIA successfully adheres to the proposed two principles, thus providing empirical support for our claims. Furthermore, under covariate shift, we also depict the stable features identified by AIA in comparison to other baseline methods (refer to Appendix E.4). It further underscores the limitations of alternative methods and highlights the superior performance of AIA.

## 5.4 Ablation Study (RQ3)

**Adversarial Augmentation & Stable Feature Learning.** As illustrated in Figure 4 (*Left*), "w/o Adv" and "w/o Sta" denote AIA without adversarial augmentation and without stable feature learning, respectively. RDIA is a variant that replaces adversarial augmentation in AIA with random augmentation (*i.e.,* random masks). The performance degrades when either component is used independently, compared to their combined application in AIA. The removal of adversarial perturbations results in a loss of the invariance condition inherent in stable feature learning [17, 29], leading to suboptimal outcomes. Conversely, the sole use of adversarial augmentation disrupts the stable features, thereby diminishing the performance. RDIA surpasses ERM, yet falls short of AIA, indicating that although randomness can foster discrepancy, it is less effective than the adversarial strategy.

**Sensitivity Analysis.** We conduct experiments to explore the sensitivities of ratio $\lambda_s$ and penalty coefficient $\gamma$. The results are displayed in Figure 4 (*Middle*) and (*Right*). $\lambda_s$ with 0.3~0.8 performs well on Motif and Molbbbp, while Molhiv is better in 0.1~0.3. It indicates that $\lambda_s$ is a dataset-sensitive hyper-parameter that needs careful tuning. For $\gamma$, the appropriate values range from 0.1~1.5.

## 6 Related Work

**Graph Data Augmentation** [22, 23, 43] enlarges the training distribution by perturbing features in graphs. Recent studies [44, 13] observe that it often outperforms other generalization efforts [14, 31]. DropEdge [25] randomly removes edges, while FLAG [24] augments node features with an adversarial strategy. M-Mixup [26] interpolates graphs in semantic space. However, studies [14, 45]

point out that stable features are the key to OOD generalization. These augmentation efforts are prone to perturb the stable features, which easily lose control of the augmented data distribution.

**Invariant Graph Learning** has been widely adopted by recent works as a paradigm for handling distribution shifts on graphs. The pioneering works [3, 17] leverage the causal invariance principle to model the invariant predictive patterns in data for the OOD generalization purpose. With the similar spirit, GREA [18] and CAL [5] aim to learn stable features by considering different environments. Some other works also utilize invariant learning to develop generalizable models and algorithms for improving the generalization *w.r.t.* molecular graphs [8] and recommender systems [9].

**Out-of-Distribution Learning on Graphs** has aroused wide research interest in the graph learning community. One line of research is centered around the goal of improving the OOD generalization capabilities of models when encountered with test data from new unseen distributions [46, 3, 47, 39, 40, 17, 38, 48, 5]. Another line of research, differently, aims to identify the OOD data in the testing set and improve the reliablity of models against OOD data for which the model should reject for prediction [49, 50, 51]. The latter task is called Out-of-Distribution Detection in the literature and serves as another under-explored area that has different technical aspect from the present work. Due to space constraints, we put more discussions of other related studies in Appendix G.

# 7 Conclusion

In this study, we address the pervasive yet largely unexplored issue of covariate shift in graph learning. We introduce a novel graph augmentation method, AIA, grounded in two principles: environmental feature discrepancy and stable feature consistency. The discrepancy principle enables the model to explore new environments, thereby facilitating better generalization to potentially unseen environments. Meanwhile, the consistency principle maintains the integrity of stable features. We conduct extensive comparisons with various baseline models and perform thorough analyses.

# 8 Limitations and Broader Impacts

This paper presents a graph augmentation method, AIA, designed to bolster the academic community's application of data augmentation methodologies. We do not foresee any immediate, direct, or adverse societal implications resulting from our study's findings. We also present additional discussions regarding AIA's limitations and potential future work in Appendix H.

## Acknowledgments and Disclosure of Funding

This research is supported by the National Natural Science Foundation of China (9227010114, U19A2079, 62302321) and the University Synergy Innovation Program of Anhui Province (GXXT-2022-040). This work is also sponsored by Ant Group through CCF-Ant Research Fund and CCF-AFSG Research Fund.

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

## A  Correlation Shift and Covariate Shift

Following graph data generation process [17, 8, 3, 29], we can observe that environmental features easily change outside the training distribution, owing to their unstable nature. Hence, distribution shifts are mainly caused by the environmental features [12, 13, 2]. Specifically, we define the joint distribution of training and test data as $P_{\text{tr}}(G,Y)$ and $P_{\text{te}}(G,Y)$, respectively. Since their joint distribution can be rewritten as $P_{\text{tr}}(G,Y) = P_{\text{tr}}(Y|G)P_{\text{tr}}(G)$ and $P_{\text{te}}(G,Y) = P_{\text{te}}(Y|G)P_{\text{te}}(G)$, we can find that there exist two main reasons that lead to the distribution shift $P_{\text{tr}}(G,Y) \neq P_{\text{te}}(G,Y)$. We give intuitive examples in Figure 5 and formal definitions of these two distribution shifts.

- **Correlation shift** $P_{\text{tr}}(Y|G) \neq P_{\text{te}}(Y|G), P_{\text{tr}}(G) = P_{\text{te}}(G)$**.** If the statistical correlation of environmental features and labels is inconsistent in training and test data, a well-fitted model in training data may fail in test data, which is also known as spurious correlation [14], correlation shift [12] or concept shift [2]. Formally, correlation shift describes the conditional distribution $P_{\text{tr}}(Y|G) \neq P_{\text{te}}(Y|G)$.

- **Covariate shift** $P_{\text{tr}}(G) \neq P_{\text{te}}(G), P_{\text{tr}}(Y|G) = P_{\text{te}}(Y|G)$**.** If there exist environmental features in the test distribution that the model has not seen during training, it will also result in a large performance drop. This unseen distribution shift is well known as covariate shift [2] or diversity shift [12]. It means that the environmental features in test data are unseen in training data, which leads to $P_{\text{tr}}(G) \neq P_{\text{te}}(G)$. In Definition 2.1, we also quantitatively measure the covariate shift between $P_{\text{tr}}(G)$ and $P_{\text{te}}(G)$.

It is worth noting that within the computer vision domain, the general covariate shift is frequently synonymous with sample selection bias [52, 53, 54, 55, 56]. Various factors contribute to covariate shift, such as heterogeneous category distribution or domain-specific variances. In the context of our investigation into graph-based models, we adhere to the assumptions outlined in previous literature [2, 12], which mainly ignore the influence of label shifts. And we posit that covariate shifts are principally attributed to the environmental features. The exploration of more comprehensive scenarios involving covariate shifts will be undertaken in our future work.

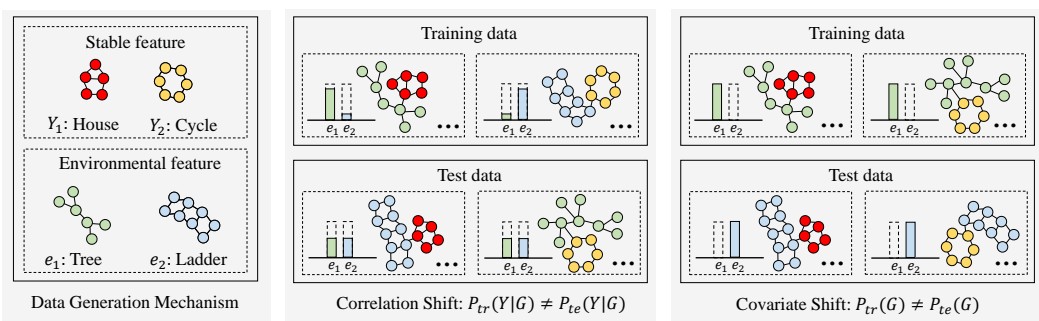

Figure 5: Intuitive examples of correlation shift and covariate shift

## B  Proofs

In this section, we provide the detailed proofs to our proposition. We start with the definition of the 1-Wasserstein distance, $D_W(P, P')$, between two distributions $P$ and $P'$:

$$D_W(P, P') = \inf_{\pi \in \Gamma(P,P')} \int_{\mathbb{R}^d \times \mathbb{R}^d} \|\boldsymbol{x} - \boldsymbol{y}\| \cdot d\pi(\boldsymbol{x}, \boldsymbol{y}),$$

where $\Gamma(P, P')$ represents the set of all joint distributions $\pi(\boldsymbol{x}, \boldsymbol{y})$ that have $P$ and $P'$ as their respective marginals. Under our conditions, due to $P'(\boldsymbol{x} - \boldsymbol{x}') = P(\boldsymbol{x})$, we can precisely determine the manner in which the mass from $P$ was transferred to create $P'$. This process involves moving each point $\boldsymbol{x}$ under $P$ to $\boldsymbol{x} + \boldsymbol{x}'$ under $P'$. Therefore, the infimum is attained by the coupling that deterministically transitions each point $\boldsymbol{x}$ to $\boldsymbol{x} + \boldsymbol{x}'$. Consequently, the Wasserstein distance simplifies to:

$$D_W(P, P') = \int_{\mathbb{R}^d} \|\boldsymbol{x} - (\boldsymbol{x} + \boldsymbol{x}')\| \cdot dP(\boldsymbol{x}) = \int_{\mathbb{R}^d} \|\boldsymbol{x}'\| \cdot dP(\boldsymbol{x}) = \|\boldsymbol{x}'\|.$$

This result establishes that the Wasserstein distance between the distributions $P$ and $P'$ is equivalent to the magnitude of $\boldsymbol{x}'$. Therefore, we obtain:

$$D_W(P, \widetilde{P}_1) = \int_\Omega \|\boldsymbol{x} - (\boldsymbol{x} + \boldsymbol{x}_1)\| \cdot d\widetilde{P}_1(\boldsymbol{x}) = \int_\Omega \|\boldsymbol{x}_1\| \cdot dP_1(\boldsymbol{x}) = \|\boldsymbol{x}_1\|.$$

Similarly, $D_W(P, \widetilde{P}_2) = \|\boldsymbol{x}_2\|$. Consequently, if $D_W(P, \widetilde{P}_1) < D_W(P, \widetilde{P}_2)$, we can deduce that $\|\boldsymbol{x}_1\| < \|\boldsymbol{x}_2\|$.

Next, we examine the GCS measure. The covariate shift between $P$ and $\widetilde{P}_1$ is given by:

$$\text{GCS}(P, \widetilde{P}_1) = \frac{1}{2} \int_{\mathcal{S}_1} |P(\boldsymbol{x}) - P(\boldsymbol{x} + \boldsymbol{x}_1)| \cdot d\boldsymbol{x}.$$

Similarly, we have:

$$\text{GCS}(P, \widetilde{P}_2) = \frac{1}{2} \int_{\mathcal{S}_2} |P(\boldsymbol{x}) - P(\boldsymbol{x} + \boldsymbol{x}_2)| \cdot d\boldsymbol{x},$$

where $\mathcal{S}_1$ and $\mathcal{S}_2$ denote the non-overlapping regions between $P$ and $\widetilde{P}_1$, and between $P$ and $\widetilde{P}_2$, respectively. Now we define another $\widetilde{P}_1'$ that satisfies $\widetilde{P}_1'(\boldsymbol{x}) = P(\boldsymbol{x} + \|\boldsymbol{x}_1\| \cdot \frac{\boldsymbol{x}_2}{\|\boldsymbol{x}_2\|})$ and let $\mathcal{S}_1'$ denote the non-overlapping regions between $P$ and $\widetilde{P}_1'$. Given the isotropy of $\widetilde{P}_1$, the integral over $\mathcal{S}_1$ is equal to the integral over $\mathcal{S}_1'$. Therefore, we can proceed to deduce:

$$\text{GCS}(P, \widetilde{P}_1) = \frac{1}{2} \int_{\mathcal{S}_1} |P(\boldsymbol{x}) - P(\boldsymbol{x} + \boldsymbol{x}_1)| \cdot d\boldsymbol{x}$$

$$= \frac{1}{2} \int_{\mathcal{S}_1'} |P(\boldsymbol{x}) - P(\boldsymbol{x} + \boldsymbol{x}_1)| \cdot d\boldsymbol{x}$$

$$= \text{GCS}(P, \widetilde{P}_1')$$

Let $\mathcal{S}_2' = \{\boldsymbol{x} + (\|\boldsymbol{x}_2\| - \|\boldsymbol{x}_1\|) \cdot \frac{\boldsymbol{x}_2}{\|\boldsymbol{x}_2\|} \mid \boldsymbol{x} \in \mathcal{S}_1'\}$, we can easily know that $|\mathcal{S}_1'| = |\mathcal{S}_2'|$ and $\mathcal{S}_2' \subseteq \mathcal{S}_2$. Thus we have:

$$\text{GCS}(P, \widetilde{P}_2) - \text{GCS}(P, \widetilde{P}_1) = \frac{1}{2} \int_{\mathcal{S}_2} |P(\boldsymbol{x}) - P(\boldsymbol{x} + \boldsymbol{x}_1)| \cdot dg - \frac{1}{2} \int_{\mathcal{S}_1} |P(\boldsymbol{x}) - P(\boldsymbol{x} + \boldsymbol{x}_1)| \cdot d\boldsymbol{x}$$

$$= \frac{1}{2} \int_{\mathcal{S}_2} |P(\boldsymbol{x}) - P(\boldsymbol{x} + \boldsymbol{x}_2)| \cdot d\boldsymbol{x} - \frac{1}{2} \int_{\mathcal{S}_1'} |P(\boldsymbol{x}) - P(\boldsymbol{x} + \boldsymbol{x}_1)| \cdot d\boldsymbol{x}$$

$$= \frac{1}{2} \int_{\mathcal{S}_2} |P(\boldsymbol{x}) - P(\boldsymbol{x} + \boldsymbol{x}_2)| \cdot d\boldsymbol{x} - \frac{1}{2} \int_{\mathcal{S}_2'} |P(\boldsymbol{x}) - P(\boldsymbol{x} + \boldsymbol{x}_2)| \cdot d\boldsymbol{x}$$

$$= \frac{1}{2} \int_{\mathcal{S}_2 \setminus \mathcal{S}_2'} |P(\boldsymbol{x}) - P(\boldsymbol{x} + \boldsymbol{x}_2)| \cdot d\boldsymbol{x} \geq 0.$$

The integral over $\mathcal{S}_2$ could not be less than the integral over $\mathcal{S}_1$, leading to $\text{GCS}(P, \widetilde{P}_1) \leq \text{GCS}(P, \widetilde{P}_2)$. Hence, we complete the proof.

## C  Estimation of Graph Covariate Shift

In this section, we elaborate on the implementation details of estimating the graph covariate shift. Without loss of generality, we start with the example of estimating the graph covariate shift between the training and test distributions. Given the training set and test set $\mathcal{D}_{\text{tr}}$ and $\mathcal{D}_{\text{te}}$, they follow probability distribution functions $P_{\text{tr}}$ and $P_{\text{te}}$. The process of estimating $\text{GCS}(P_{\text{tr}}, P_{\text{te}})$ is summarized in the following two steps:

- Firstly, it is intractable to directly estimate the distribution in graph space $\mathbb{G}$. Inspired by [12], we can obtain the graph features and estimate the distribution in feature space $\mathbb{F}$. Specifically, given a sample, we train a binary GNN classifier $f$ to distinguish which distribution it comes from, where $f(\cdot) = \Phi \circ h$, $h(\cdot) : \mathbb{G} \to \mathbb{F}$ is a graph encoder, and $\Phi(\cdot) : \mathbb{F} \to \{0, 1\}$ is a binary classifier. Then we can adopt the pre-trained GNN encoder $h$ to extract graph features.
- Secondly, we prepare the features and estimate the distribution of the data via Kernel Density Estimation (KDE) [57]. Finally, we adopt the Monte Carlo Integration under importance sampling [58] to approximate the integrals in Definition 2.1.

We summarize these implementations in Algorithm 1. In lines 4 and 5, to avoid the label shift [12], we adopt sample reweighting to ensure the balance of each class.

---

**Algorithm 1:** Estimation of Graph Covariate shift

---

**Require:** Training dataset $\mathcal{D}_{\mathrm{tr}}$ and test dataset $\mathcal{D}_{\mathrm{te}}$; Batch size $N$; Loss function $\ell$; GNN $f = \Phi \circ h$; Importance sampling size $M$; Threshold $\epsilon$.
**Ensure:** Estimated covariate shift $\mathrm{GCS}(P_{\mathrm{tr}}, P_{\mathrm{te}})$.
 1: Initialize parameters of $f$
 2:   # Train a graph classifier
 3: **while** not converge **do**
 4:     Sample a batch $\mathcal{B}_{\mathrm{tr}} \leftarrow \{(g_i, y_i)\}_{i=1}^{N} \subset \mathcal{D}_{\mathrm{tr}}$ and relabel all $y_i \leftarrow 0$
 5:     Sample a batch $\mathcal{B}_{\mathrm{te}} \leftarrow \{(g_i, y_i)\}_{i=1}^{N} \subset \mathcal{D}_{\mathrm{te}}$ and relabel all $y_i \leftarrow 1$
 6:     $\mathcal{B} \leftarrow \mathcal{B}_{\mathrm{tr}} \cup \mathcal{B}_{\mathrm{te}}$
 7:     **for** each $(g_i, y_i) \in \mathcal{B}$ **do**
 8:       Compute loss $\ell(f(g_i), y_i)$ and back-propagate gradients
 9:     **end for**
10:     Update the parameters of $f$ via gradient descent and reset the gradients
11: **end while**
12:  # Prepare the features for the estimation
13: Extract training and test feature sets $\mathcal{F}_{\mathrm{tr}}$ and $\mathcal{F}_{\mathrm{te}}$ via encoder $h$
14: $\mathcal{F} \leftarrow \mathcal{F}_{\mathrm{tr}} \cup \mathcal{F}_{\mathrm{te}}$
15: Scale $\mathcal{F}$ to zero mean and unit variance
16: $\hat{\omega} \leftarrow$ fit by KDE the distribution of $\mathcal{F}$
17: Split $\mathcal{F}$ to recover the original partition $\mathcal{F}'_{\mathrm{tr}}, \mathcal{F}'_{\mathrm{te}}$
18: $\hat{P}_{\mathrm{tr}}, \hat{P}_{\mathrm{te}} \leftarrow$ fit by KDE the distributions of $\mathcal{F}'_{\mathrm{tr}}, \mathcal{F}'_{\mathrm{te}}$
19:  # Estimate the covariate shift
20: Initialize $\mathrm{GCS}(P_{\mathrm{tr}}, P_{\mathrm{te}}) \leftarrow 0$
21: **for** $t \leftarrow \{1, ..., M\}$ **do**
22:     $z \leftarrow$ sample from $\hat{\omega}$
23:     **if** $\hat{P}_{\mathrm{tr}}(z) < \epsilon$ or $\hat{P}_{\mathrm{te}}(z) < \epsilon$ **then**
24:       $\mathrm{GCS}(P_{\mathrm{tr}}, P_{\mathrm{te}}) \leftarrow \mathrm{GCS}(P_{\mathrm{tr}}, P_{\mathrm{te}}) + |\hat{P}_{\mathrm{tr}}(z) - \hat{P}_{\mathrm{te}}(z)|/\hat{\omega}(z)$
25:     **end if**
26: **end for**
27: $\mathrm{GCS}(P_{\mathrm{tr}}, P_{\mathrm{te}}) \leftarrow \mathrm{GCS}(P_{\mathrm{tr}}, P_{\mathrm{te}})/2M$

---

**Algorithm 2:** Adversarial Invariant Augmentation

---

**Require:** Training set $\mathcal{D}_{\mathrm{tr}}$; Adversarial augmenter $T_{\theta_1}(\cdot)$; Stable feature generator $T_{\theta_2}(\cdot)$; GNN classifier $f(\cdot)$ with parameters $\theta$; Learning rates $\alpha, \beta$; Batch size $N$; Stable feature ratio $\lambda_s$; Penalty $\gamma$.
 1: Randomly initilize $\theta, \theta_1, \theta_2$
 2: **while** not converge **do**
 3:     Sample a batch $\mathcal{B}_{\mathrm{tr}} \leftarrow \{(g_i, y_i)\}_{i=1}^{N} \subset \mathcal{D}_{\mathrm{tr}}$
 4:     **for** each $(g_i, y_i) \in \mathcal{B}_{\mathrm{tr}}$ **do**
 5:       $\mathbf{M}_{\mathrm{adv}}^a, \mathbf{M}_{\mathrm{adv}}^x \leftarrow T_{\theta_1}(g_i)$ `// adversarial perturbations`
 6:       $\mathbf{M}_{\mathrm{sta}}^a, \mathbf{M}_{\mathrm{sta}}^x \leftarrow T_{\theta_2}(g_i)$ `// regions of stable features`
 7:       $\widetilde{\mathbf{M}}^a \leftarrow (1 - \mathbf{M}_{\mathrm{sta}}^a) \odot \mathbf{M}_{\mathrm{adv}}^a + \mathbf{M}_{\mathrm{sta}}^a$ `// augment edges`
 8:       $\widetilde{\mathbf{M}}^x \leftarrow (1 - \mathbf{M}_{\mathrm{sta}}^x) \odot \mathbf{M}_{\mathrm{adv}}^x + \mathbf{M}_{\mathrm{sta}}^x$ `// augment nodes`
 9:       $\widetilde{g_i} \leftarrow (\mathbf{A}_i \odot \widetilde{\mathbf{M}}^a, \mathbf{X}_i \odot \widetilde{\mathbf{M}}^x)$ `// augmented graph`
10:     **end for**
11:     Compute $\mathcal{L}_{\mathrm{adv}} - \mathcal{L}_{\mathrm{reg}_1}$ via Equation (7) and (9)
12:     Compute $\mathcal{L}_{\mathrm{sta}} + \mathcal{L}_{\mathrm{reg}_2}$ via Equation (8) and (10)
13:     Update parameters of adversarial augmenter via gradient ascent:
       $\theta_1 \leftarrow \theta_1 + \alpha \nabla_{\theta_1}(\mathcal{L}_{\mathrm{adv}} - \mathcal{L}_{\mathrm{reg}_1})$
14:     Update parameters of GNN and stable feature generator via gradient descent:
       $\theta \leftarrow \theta - \beta \nabla_{\theta}(\mathcal{L}_{\mathrm{sta}} + \mathcal{L}_{\mathrm{reg}_2}); \theta_2 \leftarrow \theta_2 - \beta \nabla_{\theta_2}(\mathcal{L}_{\mathrm{sta}} + \mathcal{L}_{\mathrm{reg}_2})$
15: **end while**

---

Table 4: Statistics of graph classification datasets.

| Dataset | | Motif | | CMNIST | Molbbbp | | Molhiv | |
|---|---|---|---|---|---|---|---|---|
| Covariate shift | | base | size | color | scaffold | size | scaffold | size |
| Train | Graph# | 18000 | 18000 | 42000 | 1631 | 1633 | 24682 | 26169 |
| | Avg. node# | 17.07 | 16.93 | 75.00 | 22.49 | 27.02 | 26.25 | 27.87 |
| | Avg. edge# | 48.89 | 43.57 | 1392.76 | 48.43 | 58.71 | 56.68 | 60.20 |
| Val | Graph# | 3000 | 3000 | 7000 | 204 | 203 | 4113 | 2773 |
| | Avg. node# | 15.82 | 39.22 | 75.00 | 33.20 | 12.06 | 24.95 | 15.55 |
| | Avg. edge# | 33.00 | 107.03 | 1393.73 | 71.84 | 24.27 | 54.53 | 32.77 |
| Test | Graph# | 3000 | 3000 | 7000 | 204 | 203 | 4108 | 3961 |
| | Avg. node# | 14.97 | 87.18 | 75.00 | 27.51 | 12.26 | 19.76 | 12.09 |
| | Avg. edge# | 31.54 | 239.65 | 1393.60 | 59.75 | 24.87 | 40.58 | 24.87 |
| Class# | | 3 | 3 | 10 | 2 | 2 | 2 | 2 |

Table 5: Hyper-parameter details of AIA.

| Dataset | Motif | | CMNIST | Molbbbp | | Molhiv | |
|---|---|---|---|---|---|---|---|
| Covariate shift | base | size | color | scaffold | size | scaffold | size |
| Backbone (layer-hidden) | 4-300 | 4-300 | 4-300 | 4-64 | 4-32 | 4-128 | 4-128 |
| Augmenter (layer-hidden) | 2-300 | 2-300 | 2-300 | 2-64 | 2-32 | 2-128 | 2-128 |
| Generator (layer-hidden) | 2-300 | 2-300 | 2-300 | 2-64 | 2-32 | 2-128 | 2-128 |
| Optimizer | Adam | Adam | Adam | Adam | Adam | Adam | Adam |
| Learning rate $\alpha$ | 1e-3 | 1e-3 | 1e-3 | 1e-3 | 5e-3 | 1e-3 | 1e-2 |
| Learning rate $\beta$ | 5e-3 | 5e-3 | 5e-3 | 1e-3 | 5e-3 | 1e-2 | 1e-2 |
| Stable feature ratio $\lambda_s$ | 0.5 | 0.5 | 0.5 | 0.5 | 0.5 | 0.1 | 0.1 |
| Adversarial penalty $\gamma$ | 0.2 | 0.2 | 0.2 | 0.5 | 0.5 | 0.5 | 0.5 |

# D  Implementation Details

## D.1  Algorithm

We summarize the detailed implementations of AIA in Algorithm 2. We alternately optimize the adversarial augmenter and stable feature generator with the backbone model, in lines 13 and 14. We adopt the learned stable features for predictions in the inference stage.

## D.2  Datasets

In this paper, we conduct experiments on graph OOD datasets [2] and OGB datasets [20], which include Motif, CMNIST, Molbbbp and Molhiv. We follow [2] to create various covariate shifts, according to base, color, size and scaffold splitting. Base, color, size and scaffold are features of the graph data and do not determine the labels of the data, so they can be regarded as environmental features. The statistics of the datasets are summarized in Table 4. Below we give a brief introduction to each dataset.

- **Motif:** It is a synthetic dataset from Spurious-Motif [17, 5]. As shown in original graphs in Figure 5, each graph is composed of a base-graph (*wheel, tree, ladder, star, path*) and a motif (*house, cycle, crane*). The label is only determined by the type of motif. We create covariate shift according to the base-graph type and the graph size (*i.e.,* node number). For base covariate shift, we adopt graphs with *wheel, tree, ladder* base-graphs for training, *star* for validation and *path* for testing. For size covariate shift, we use small-size of graphs for training, while the validation and the test sets include the middle- and the large-size graphs, respectively.
- **CMNIST:** Color MNIST dataset contains graphs transformed from MNIST via superpixel techniques [59]. We define color as the environmental features to create the covariate shift. Specifically, we color digits with 7 different colors, where five of them are adopted for training while the remaining two are used for validation and testing.
- **Molbbbp & Molhiv:** These are molecular datasets collected from MoleculeNet [19]. We define the scaffold and graph size (*i.e.,* node number) as the environmental features to create two types of covariate shifts. For scaffold shift, we follow [2] and use scaffold split to create training, validation

Table 6: Performance over diverse backbones.

| Backbone | Method | Motif | | Molbbbp | |
| --- | --- | --- | --- | --- | --- |
| | | base | size | scaffold | size |
| GCN | EEM | $67.41_{\pm4.47}$ | $50.76_{\pm2.92}$ | $66.44_{\pm1.91}$ | $77.79_{\pm4.04}$ |
| | M-Mixup | $68.83_{\pm4.04}$ | $51.50_{\pm4.95}$ | $67.09_{\pm0.57}$ | $78.42_{\pm2.71}$ |
| | FLAG | $59.87_{\pm5.61}$ | $50.99_{\pm4.18}$ | $66.03_{\pm2.59}$ | $78.76_{\pm2.54}$ |
| | AIA (ours) | $\mathbf{72.39_{\pm5.37}}$ | $\mathbf{54.87_{\pm5.02}}$ | $\mathbf{69.13_{\pm1.76}}$ | $\mathbf{80.53_{\pm4.43}}$ |
| GCNII | EEM | $67.84_{\pm4.74}$ | $51.74_{\pm3.15}$ | $67.64_{\pm1.84}$ | $77.96_{\pm4.02}$ |
| | M-Mixup | $67.53_{\pm4.31}$ | $52.31_{\pm5.18}$ | $66.64_{\pm0.50}$ | $76.67_{\pm2.69}$ |
| | FLAG | $58.67_{\pm5.88}$ | $50.18_{\pm4.41}$ | $66.72_{\pm2.52}$ | $78.55_{\pm2.52}$ |
| | AIA (ours) | $\mathbf{72.70_{\pm4.14}}$ | $\mathbf{53.23_{\pm6.25}}$ | $\mathbf{67.93_{\pm1.69}}$ | $\mathbf{79.72_{\pm3.37}}$ |
| GAT | EEM | $66.53_{\pm4.42}$ | $51.16_{\pm3.22}$ | $66.51_{\pm1.77}$ | $77.41_{\pm3.81}$ |
| | M-Mixup | $69.25_{\pm3.99}$ | $51.37_{\pm5.25}$ | $66.95_{\pm0.43}$ | $77.21_{\pm2.48}$ |
| | FLAG | $59.53_{\pm5.56}$ | $51.32_{\pm4.48}$ | $67.36_{\pm2.45}$ | $77.87_{\pm2.31}$ |
| | AIA (ours) | $\mathbf{71.95_{\pm3.39}}$ | $\mathbf{54.38_{\pm6.32}}$ | $\mathbf{69.21_{\pm1.62}}$ | $\mathbf{78.49_{\pm3.20}}$ |

and test sets. For size shift, we adopt the large-size of graphs for training and the smaller ones for validation and testing.

### D.3 Metrics

We adopt classification accuracy as the metric for Motif and CMNIST. As suggested by [20], we use ROC-AUC for Molhiv and Molbbbp datasets. In addition, we use $\mathrm{GCS}(P, Q)$ to measure the covariate shift between distributions $P$ and $Q$. For all experimental results, we perform 10 random runs and report the mean and standard derivations. For augmentation diversity, we use conditional entropy to measure the diversity of generated data. We normalize it to $[0, 1]$ for better comparison. Specifically, for a given graph data, we compute the conditional entropy by collecting augmented data generated during the training process. We conduct experiments by collecting 1000 graph data, and report the mean and standard deviation.

### D.4 Training Settings

We use the NVIDIA GeForce RTX 3090 (24GB GPU) to conduct all our experiments. To make a fair comparison, we adopt GIN [60] as the default architecture to conduct all experiments. We tune the hyper-parameters in the following ranges: $\alpha$ and $\beta \in \{0.01, 0.005, 0.001\}$; $\lambda_2 \in \{0.1, ..., 0.9\}$; $\gamma \in \{0.01, 0.1, 0.2, 0.5, 1.0, 1.5, 2.0, 3.0, 5.0\}$. The hyper-parameters are summarized in Table 5.

### D.5 Baseline Settings

For a more comprehensive comparison, we selected 16 baselines. In this section, we give a detailed introduction to the settings of these methods.

- For ERM, IRM [14], GroupDRO [31], VREx [37], and M-Mixup [26], we report the results from the study [2] by default and reproduce the missing results on Molbbbp.

- For DIR [17], CAL [5], GSAT [38], DropEdge [25], GREA [18], FLAG [24], $\mathcal{G}$-Mixup [7], CIGA [41] and DisC [21], they provide source codes for the implementations. We adopt default settings from their source codes and detailed hyper-parameters from their original papers.

- For OOD-GNN [39] and StableGNN [40], their source codes are not publicly available. We reproduce them based on the codes of StableNet [16].

- For RDIA in Section 5.4, it is a variant that replaces the adversarial augmentation in AIA with random augmentation. In our implementation, we use all-one matrices to create the initial node and edges masks. Then we randomly set 20% of nonzero elements to zero in these masks. Finally, we apply these masks to the graphs for random data augmentation. The process of stable feature learning is consistent with AIA.

## E  More Experimental Results

### E.1  Results on Correlation Shift

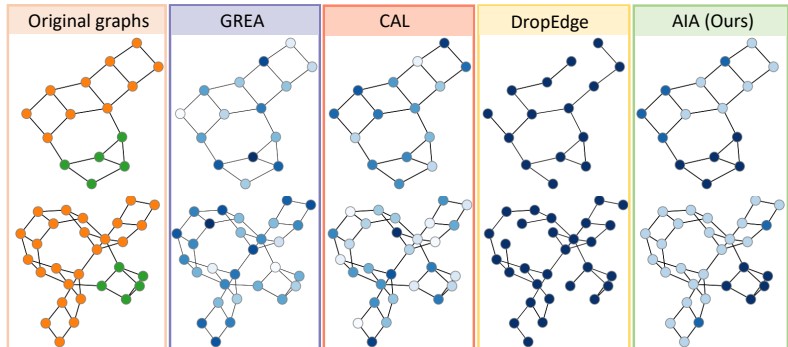

Figure 6: Visualizations of captured stable features.

Although this work focuses on the OOD issue of covariate shift, for completeness, we also evaluate the performance of AIA under correlation shift. Following [2], we choose three graph OOD datasets (*i.e.,* Motif, CMNIST, Molhiv) with three different graph features (*i.e.,* base, color, size) to create correlation shifts. For baselines, we choose three generalization algorithms (*i.e.,* ERM, IRM [14], VREx [37]), three graph generalization methods (*i.e.,* DIR [17], CAL [5], OOD-GNN [39]) and three data augmentation methods (*i.e.,* DropEdge [25], FLAG [24], M-Mixup [26]). The experimental results are shown in Table 7. We can observe that AIA can also

Table 7: Performance on correlation shift.

| Method | Motif | CMNIST | Molhiv |
|---|---|---|---|
| ERM | $81.44_{\pm2.54}$ | $42.87_{\pm1.37}$ | $63.26_{\pm1.25}$ |
| IRM | $80.71_{\pm2.81}$ | $42.80_{\pm1.62}$ | $59.90_{\pm1.17}$ |
| VREx | $81.56_{\pm2.14}$ | $43.31_{\pm1.03}$ | $60.23_{\pm1.60}$ |
| DIR | $73.25_{\pm6.37}$ | $38.78_{\pm1.45}$ | $66.78_{\pm1.50}$ |
| CAL | $81.94_{\pm1.20}$ | $41.82_{\pm0.85}$ | $62.36_{\pm1.42}$ |
| OOD-GNN | $80.22_{\pm2.28}$ | $39.03_{\pm1.24}$ | $57.49_{\pm1.08}$ |
| DropEdge | $78.97_{\pm3.41}$ | $38.43_{\pm1.94}$ | $54.92_{\pm1.73}$ |
| FLAG | $80.91_{\pm1.04}$ | $43.41_{\pm1.38}$ | $66.44_{\pm2.32}$ |
| M-Mixup | $77.63_{\pm1.12}$ | $40.96_{\pm1.21}$ | $64.87_{\pm1.36}$ |
| AIA (ours) | $\mathbf{82.51_{\pm2.81}}$ | $\mathbf{49.73_{\pm1.70}}$ | $\mathbf{68.11_{\pm1.82}}$ |

effectively alleviate the correlation shift. These results demonstrate that AIA learns better stable features by encouraging environmental discrepancy, which can effectively break spurious correlations that are hidden in the training data.

### E.2 Results on Diverse Backbones

We select three different GNN backbone models (GCN [61], GCNII [62] and GAT [63]) for experiments. From the results in Table 6, our observations and conclusions remain the same with the diverse backbones.

### E.3 Results on More Real-world Datasets

To demonstrate the effectiveness of the proposed AIA, we also conduct experiments on commonly used TU datasets [64], which include MUTAG, NCI1, PROTEINS, COLLAB, IMDB-B, IMDB-M. For training settings, we follow CAL [5] and adopt GIN [60] as our backbone model. The experimental results are shown in Table 8. For the results, we can observe that our method can achieve the best performance over different datasets.

### E.4 More Visualizations

To demonstrate the superiority of our method, we also visualize the captured stable features by AIA and compare them with other baselines. The results are displayed in Figure 6. From the results, we can easily observe that our method can find stable parts more accurately than other baseline methods.

## F Complexity Analyses

Firstly, we define the average numbers of nodes and edges per graph in the dataset to be $n$ and $m$, respectively. Let $N$ denote the batch size, $l$, $l_a$ and $l_c$ denote the numbers of layers in the GNN backbone, adversarial augmenter and stable feature generator, respectively. $d$, $d_a$ and $d_c$ are the

Table 8: Performance comparisons on TU datasets.

| Method | MUTAG | NCI1 | PROTEINS | COLLAB | IMDB-B | IMDB-M |
|---|---|---|---|---|---|---|
| ERM | $89.42_{\pm7.40}$ | $82.71_{\pm1.52}$ | $76.21_{\pm3.83}$ | $82.08_{\pm1.51}$ | $73.40_{\pm3.78}$ | $51.53_{\pm2.97}$ |
| CAL | $89.91_{\pm8.34}$ | $83.89_{\pm1.93}$ | $76.92_{\pm3.31}$ | $82.68_{\pm1.25}$ | $74.13_{\pm5.21}$ | $52.60_{\pm2.36}$ |
| DropEdge | $86.11_{\pm9.41}$ | $82.35_{\pm3.77}$ | $74.40_{\pm3.10}$ | $80.59_{\pm2.14}$ | $72.34_{\pm5.83}$ | $51.06_{\pm3.04}$ |
| FLAG | $89.45_{\pm7.20}$ | $82.67_{\pm2.12}$ | $76.89_{\pm3.66}$ | $82.48_{\pm1.79}$ | $73.37_{\pm4.94}$ | $52.16_{\pm2.70}$ |
| M-Mixup | $89.83_{\pm7.67}$ | $83.89_{\pm2.38}$ | $76.76_{\pm3.40}$ | $82.90_{\pm1.43}$ | $74.07_{\pm4.76}$ | $52.89_{\pm2.84}$ |
| AIA (ours) | $\mathbf{90.34_{\pm7.75}}$ | $\mathbf{84.12_{\pm2.64}}$ | $\mathbf{77.92_{\pm3.72}}$ | $\mathbf{82.98_{\pm1.76}}$ | $\mathbf{74.23_{\pm5.10}}$ | $\mathbf{53.02_{\pm2.76}}$ |

Table 9: Running time, model size and performance improvement.

| Dataset | ERM | | CAL | | | DIR | | | AIA (ours) | | |
|---|---|---|---|---|---|---|---|---|---|---|---|
| | Running Time | Model Size | Running Time | Model Size | Performance Improvement | Running Time | Model Size | Performance Improvement | Running Time | Model Size | Performance Improvement |
| Motif | 00h 51m 19s | 1.515M | 01h 37m 15s | 2.213M | ↓ 2.98% | 01h 50m 27s | 2.158M | ↓ 5.03% | 01h 49m 08s | 2.366M | ↑ 7.55% |
| CMNIST | 01h 56m 32s | 1.517M | 02h 28m 49s | 2.244M | ↓ 2.13% | 02h 34m 35s | 2.161M | ↑ 16.08% | 02h 45m 30s | 2.373M | ↑ 27.17% |
| Molbbbp | 00h 11m 58s | 1.515M | 00h 18m 22s | 2.213M | ↑ 0.81% | 00h 20m 11s | 2.158M | ↓ 2.12% | 00h 19m 37s | 2.366M | ↑ 3.72% |
| Molhiv | 00h 27m 19s | 1.515M | 00h 46m 14s | 2.213M | ↓ 3.23% | 00h 58m 40s | 2.158M | ↓ 2.59% | 00h 55m 46s | 2.366M | ↑ 2.54% |

dimensions of hidden layers in the GNN backbone, adversarial augmenter and stable feature generator, respectively.

**Time Complexity.** The time complexity of the adversarial learning objective is $\mathcal{O}(N(l_a m d_a + 2lmd))$. For the stable feature learning objective, the time complexity is $\mathcal{O}(N(l_c m d_c + 2lmd))$. For the regularization terms, the time complexity is $\mathcal{O}(2N(n+m))$. For simplicity, we assume $l_a = l_c$ and $d_a = d_c$. Hence, the time complexity of a forward propagation is $\mathcal{O}(2N(l_a m d_a + 2lmd + n + m))$.

**Model Size.** In addition to the GNN backbone model, we also introduce two small networks for adversarial augmentation and stable feature learning. In our implementations, the parameters of AIA are around twice as large as those of the original GNN model. The running time and model size are shown in Table 9.

## G  More Related Studies

**OOD Generalization** [1, 65, 66] has been widely explored. Inspired by invariant learning and causal theory, a series of general algorithms [67, 68, 69, 66, 70] have recently been proposed to solve the OOD problem. Recent studies [12, 2, 13] point out that OOD falls into two specific categories: correlation shift (*aka.* concept shift) and covariate shift (*aka.* diversity shift). Correlation shift denotes that the environmental features and labels establish a statistical correlation that is inconsistent in training and test data. Thus, the models prefer to learn spurious correlations and rely on shortcut features [71] for predictions, resulting in a large performance drop. In contrast, covariate shift indicates that there exist unseen environmental features in test data. The limited training environment makes this issue intractable. Tasks pertaining to domain generalization [72] often exhibit the covariate shift issue, such as model inference under previously unseen test domains. Consequently, there have been numerous recent efforts [73, 74, 75, 76] to address such challenges. In recent years, OOD generalization on graphs is drawing widespread attention [4]. Given the intricate nature of graph data types and associated tasks [77], issues related to distribution shifts can emerge in various contexts, such as node classification [46, 3, 47], graph classification [39, 40, 17, 38, 48, 5, 18, 78, 79], or dynamic graph data [80], among others. Emerging research has increasingly focused on the identification of stable features or subgraphs within graph data. These substructures are posited to maintain causal relationships with target labels, thereby enhancing the model's capacity for generalization [5, 18], explainability [81, 17, 38, 82], and computational efficiency [83, 84, 85]. Concurrently, efforts are being made to extend these principles to diverse applications, such as molecular graphs [8], recommender systems [9, 10], and anomaly detection [11, 86, 87].

**Comprehensive Comparisons with EERM** [3]. Although EERM shares similar goals with us, generating several environments through augmentation, there exist many technical and contribution differences. Firstly, EERM ignores the distinction between correlation shift and covariate shift problems, while we distinguish these two shifts in detail and design a framework specifically for covariate shift. Secondly, EERM does not model stable and environmental features, which results in the inability to explicitly distinguish them. In contrast, we explicitly model the environmental and stable features. Hence, we can effectively identify stable and environmental features and explicitly

Table 10: Comparisons with EERM.

| | | EERM | Our AIA |
|---|---|---|---|
| Scope | Is it specifically designed for covariate shift? | ✗ | ✓ |
| Separability | Can environmental/stable features be separated? | ✗ | ✓ |
| Environmental Feature Discrepancy | Can environmental features be identified explicitly? | ✗ | ✓ |
| | How to model environmental features? | - | Mask model $T_{\theta_1}(\cdot)$ |
| | Metric for environmental Discrepancy | - | $\text{GCS}(\widetilde{P}, P)$ |
| | Generation principle for environmental features | "Blindly" maximize $\mathbb{V}_e[R(e)]$ | Maximize $\text{GCS}(\widetilde{P}, P)$ |
| Stable Feature Consistency | Can stable features be identified explicitly? | ✗ | ✓ |
| | How to model stable features? | - | Mask model $T_{\theta_2}(\cdot)$ |
| | Learning principles for stable features | $\min_\theta \mathbb{V}_e[R(e)]$ | Sufficiency/Independence |
| Generalization | Theoretical basis | IRM | DRO |
| | Generalization scope | $K$ environments | Robust radius $\rho$ $D(\widetilde{P}, P) \le \rho$ |

separate them from data. Thirdly, we also design a metric, $\text{GCS}(\widetilde{P}, P)$, which can effectively measure the discrepancy of the environmental features for our augmented data. And we directly encourage the environmental discrepancy of the augmented samples by maximizing $\text{GCS}(\widetilde{P}, P)$. However, EERM does not provide any evaluation metric for environmental discrepancy. To encourage the discrepancy, they "blindly" maximize the variance of the empirical risk in $K$ environments. Finally, for generalization scope, EERM is based on the IRM [14] by minimizing the empirical risk in $K$ environments. In contrast, inspired by DRO [31], we can guarantee the generalization within the robust radius $\rho$. We summarize the above detailed discussions in Table 10.

# H   Limitation & Future Work

Although AIA outperforms numerous baselines and can achieve outstanding performance under various covariate shifts, we also prudently introspect the following limitations of our method. And we leave the improvements of these limitations as our future work.

- AIA performs OOD exploration through an adversarial data augmentation strategy to achieve environmental discrepancy. However, it only perturbs the existing graph data in a given training set, such as perturbing original graph node features or graph structures. Hence, it is possible that there still exist some overlaps between the augmented distribution and training distribution, so discrepancy principle cannot be thoroughly achieved. In future work, we will attempt to design more advanced data augmentation methods, such as graph generation-based strategies [88], to generate more unseen and novel graph data, for pursuing the discrepancy principle.

- For model training, we adopt adversarial training and stable feature learning to alternately optimize the adversarial augmenter, stable feature generator and backbone GNN. This training strategy may make the training process unstable, so the performance of AIA may experience a large variance. In addition, these two networks also involve additional parameters. Optimizing these parameters separately will also increase the time complexity, as shown in Appendix F. Hence, in future work, we will explore how to utilize more advanced optimization methods and lightweight models to achieve the principles of environmental feature discrepancy and stable feature consistency.

