# OpenReview forum: "Unleashing the Power of Graph Data Augmentation on Covariate Distribution Shift"
_NeurIPS.cc/2023/Conference — NeurIPS 2023 poster_

### Official Review · Reviewer_J9ZS · 2023-07-07

**Soundness:** 3 good
**Presentation:** 3 good
**Contribution:** 2 fair
**Rating:** 6
**Confidence:** 4

**Summary:**

This paper proposes an augmentation technique to handle covariate shift during graph classification by creating new "environmental factors" and simultaneously preserving "stable" factors. Since unseen environmental factors occur during covariate shift, the authors propose using an adversarial augmentor to find augmentations that increase the "GCS." However, since unconstrained adversarial augmentation could harm the "stable" factors, the method includes a second network that learns to a mask over the stable nodes/edges. The found mask is used to help keep the adversarial augmentation to environmental factors and also in the loss. The method is shown to do well on the a variety of benchmarks.

**POST REBUTTAL:** I have read the other reviews and the author's rebuttals. I thank the authors for their thorough responses! The explanations and additional experiments (non-random augs., graph transformers) are helpful and strengthen the paper. I encourage the authors to add these discussions, and experiments to their final paper as well as cite the suggested related work on task-relevant invariances. My concerns have been addressed and have raised the score!

**Strengths:**

- Motivation: The method is well-motivated: the environmental feature discrepancy and stable feature consistency objectives for promoting performance under covariate shift make sense.

- Strong experimental results. The proposed method is evaluated against many baselines on several different benchmarks. The evaluation of augmentation diversity and discrepancy is also useful.



**Weaknesses:**

- Limited Novelty: Individual pieces, as well as the motivation for the method are not particularly novel. These concepts have been discussed in the context of designing augmentations in general and also in the context of creating augmentations for SSL (in-variance/recover-ability, style/content). Adversarial augmentation has been explored by other methods, and masking for explanation too. To this end, I don't think that the theoretical discussion adds too much.

- Complexity: Including two networks, as well as an adversarial objective is expensive and potentially more difficult to train. Moreover, the masking operation can be expensive, since the mask must be learned over all the nodes and the edges.


**Questions:**

- The discussion of augmentation diversity in Table 3 is interesting. However, I was wondering if it would be possible to also compare to one of the augmentation methods too that is not random? Curious to see if how well something like FLAG compares.

- Quick (not important) question: how where hyper-parameters selected during tuning?

- Quick question: can you add some run-times/model sizes to the appendix for later drafts?

**Limitations:**

See above. Negative societal impacts, etc are adequately discussed.

---

> ### Author Rebuttal · Authors · 2023-08-09
>
> Thanks for your valuable time and comments! We provide detailed responses as below, and some necessary results are in our **one-page PDF** in the Global Response above. We sincerely hope that addressing your concerns will help change your rating of our work!
> ### Q1. Limited Novelty & Motivation are not novel & Adversarial augmentation has been explored by other methods.
> We appreciate your thoughts but respectfully argue that our method and motivation are novel. Now we list the following aspects.
> - **Reviewer Feedback**. The novelty and motivation have been praised by other three reviewers, such as ***Reviewer ExMB***'s comment: "The adversarial augmentation method design is novel with good intuitions and it provides new insights into the important OOD area of graph learning."; ***Reviewer bjaC***:"The technique proposed by the authors is interesting, and the use of two components on graph data, is new."; and ***Reviewer reTY***: "I believe the extension and its implementation to this problem is sufficiently novel."
> - **Our Novelty & Motivation.** Covariate shift on graphs remains largely under-explored (comments from ***Reviewer ExMB***). In this paper, we provide a clear formulation for this new problem, design a new method to solve this problem with solid theoretical analyses. Further, we also propose a new metric, to measure the graph covariate shift in diverse datasets (see Table 2 in our paper). Moreover, using invariance principle with graph augmentation for handling graph covariate shift is new and unique.
> - **Comparisons with Similar Methods (refer to our one-page PDF).** We understand your concern that our augmentation is not novel. However, most adversarial augmentation strategies [1-5] are performed on European data (e.g. image). Due to the non-Euclidean nature of graph data, it is difficult to directly transfer these strategies to graphs. Therefore, it is non-trivial to design new method in graph domain (comments from ***Reviewer bjaC***). There are few studies applying adversarial augmentation or masking to graph data. ***In our one-page PDF (Table 3)***, we list several graph augmentation methods (ADGCL [6], RGCL [7], EERM, FLAG, GREA) that are most similar to ours, and highlight the main differences.
> ### Q2. Table 3 (augmentation diversity) is interesting & Compare non-random augmentation methods (e.g., FLAG).
> We compare with non-random methods such as FLAG, G-Mixup, ADGCL, RGCL. Since ADGCL and RGCL are SSL settings, we replace our augmentation module in AIA with theirs. From the results ***in our one-page PDF (Table 4)***, FLAG mainly focuses on node feature, which limits its diversity. And FLAG does not distinguish between stable and environmental features. For G-Mixup, despite having high environmental diversity, struggles to maintain stable feature consistency. Others like ADGCL and RGCL also fall short in comparison to our performance.
> ### Q3. How to select hyper-parameters during tuning?
> We provide detailed hyper-parameters in Appendix (Table 2). They are tuned within the following ranges: $\alpha, \beta \in \\{0.01, 0.005, 0.001\\}$; $\lambda_s \in \\{0.1, ..., 0.9\\}$; $\gamma \in \\{0.01, 0.1, 0.2, 0.5, 1.0, 1.5, 2.0, 3.0, 5.0\\}$. The best parameters are chosen based on accuracy on the validation set. For basic settings like batch size, optimizer, we keep them consistent with baseline models.
> ### Q4. Complexity & Adversarial objective and masking operation is expensive. & Can you add some run-times/model sizes to the appendix for later drafts?
> In Appendix F, we have already discussed our time complexity and model size. Here we discuss more details about complexity, model size and running time. We would also be happy to include these details in our final version if the paper is accepted.
> - **Adversarial objective complexity .** In our training steps, a single step of updating the parameters of the adversarial generator can achieve good performance, instead of using a multi-step approach similar to PGD. So our complexity is acceptable.
> - **Masking operation complexity.**  Rather than computing the scores between all pairs of nodes (i.e., $O(n^2)$). We predict the score for each edge. Therefore, our time complexity is $O(m)$. For node feature, the time complexity is $O(n)$, so the total complexity for masking operation is $O(n+m)$. Furthermore, we focus on graph classification. In OGB dataset, the largest dataset has an average of only 243 nodes and 2261 edges. Hence, our approach is suitable to handle them and will not include more time complexity.
> - **Running time and Model size.** ***In our one-page PDF (Table 5)***, we count the running time and model size of ERM, DIR, CAL and our AIA on NVIDIA 3090. Our running time is about 1.5-2 times that of the base model. With two additional small networks, our model size is roughly 1.5 times the size of the original model. And our method is comparable to the current SOTA methods DIR, CAL in terms of running time and model size. Hence, we believe we achieve a better performance-complexity trade-off considering that ours can achieve significant accuracy gains. In practical applications, we believe that these additional complexities are acceptable. Of course, **we accept your suggestion and will put these discussions and results in our paper**. And we will also find ways to reduce our complexity in future work.
>
> [1] Adversarial AutoAugment, ICLR 2020
>
> [2] Maximum-Entropy Adversarial Data Augmentation for Improved Generalization and Robustness, NeurIPS 2020
>
> [3] AugMax: Adversarial Composition of Random Augmentations for Robust Training, NeurIPS 2021
>
> [4] Rethinking the Effect of Data Augmentation in Adversarial Contrastive Learning, ICLR 2023
>
> [5] Harnessing OOD Examples via Augmenting Content and Style, ICLR 2023
>
> [6] Adversarial Graph Augmentation to Improve Graph Contrastive Learning, NeurIPS 2021
>
> [7] Let Invariant Rationale Discovery Inspire Graph Contrastive Learning, ICML 2022

---

### Official Review · Reviewer_ExMB · 2023-07-07

**Soundness:** 3 good
**Presentation:** 4 excellent
**Contribution:** 3 good
**Rating:** 6
**Confidence:** 2

**Summary:**

The paper studies on the covariate shift problem in graph classification tasks, as opposed to the more commonly studied correlation shift. The authors propose a novel data augmentation strategy, Adversarial Invariant Augmentation (AIA), guided by two principles, environmental feature discrepancy and stable feature consistency. The method leverages an adversarial augmenter to adversarial generate masks. Another stable feature generator is also utilized to promote stable feature consistency. The proposed approach equips the graph classifier model with an enhanced ability to identify stable features in new environments and effectively mitigate the covariate shift issue.

**Post rebuttal**: I'm content with the rebuttal of the authors and will keep my rating.

**Strengths:**

- This paper addresses the under-explored perspective on graph OOD, which is covariate shift. It provides new insights into the important OOD area of graph learning.
- This paper designs the AIA methodology corresponding with the two proposed principles. The adversarial augmentation method design is novel with good intuitions.
- The experimental results are extensive covering a list of baseline methods and datasets with diverse properties. AIA outperforms specialized graph generalization and augmentation algorithms.
- The authors provide in-depth analysis discussing experiments with respect to the principles, which is persuasive.
- Ablation study demonstrates the effectiveness of the designed components.
- The structure of the paper is clear. The writing is easy to follow.

**Weaknesses:**

- Advsersarial training is notoriously inefficient at neural network training, so leveraging adversarial training to solve the graph OOD problem may affect the training/convergence speed of GNNs. It would be better if authors can provide discussions regarding such overhead.

**Questions:**

- What is the GNN backbone used in AIA for the experiments?
- How will the choice of GNN backbones affect the final outcome?

**Limitations:**

The paper addresses the limitations in the appendix.

---

> ### Author Rebuttal · Authors · 2023-08-09
>
> We gratefully thank you for the positive feedback and constructive comments! To address your concerns, we provide point-to-point responses, and some necessary results are in our **one-page PDF** in the Global Response above.
>
> ### Q1. Adversarial training is inefficient.
>
> In Appendix F (from Supplementary Material), we have already discussed our optimization complexity and model size. In our training steps, a single step of updating the parameters of the adversarial generator can achieve good performance, instead of using a multi-step approach similar to PGD. So our complexity is acceptable. ***In our one-page PDF (Table 5)***, we count the running time and model size of ERM, DIR [1], CAL [2] and our AIA on NVIDIA 3090 (24GB GPU). Our running time is about 1.5-2 times that of the base model. With two additional small networks, our model size is roughly 1.5 times the size of the original model. And our method is comparable to the current SOTA methods DIR, CAL in terms of running time and model size. Hence, we believe we achieve a better performance-complexity trade-off considering that ours can achieve significant performance improvements. In practical applications, we believe that these additional complexities are acceptable. Of course, we accept your suggestion and will put these discussions and results in our final version. And we will also find ways to reduce our complexity in our future work.
>
> ### Q2. What is the GNN backbone used in AIA for the experiments?
>
> For a fair comparison, we uniformly choose GIN [3] as the backbone for all algorithms.
>
> ### Q3. How will the choice of GNN backbones affect the final outcome?
>
> ***In our one-page PDF (Table 2)***, we also selected three different backbone models (GCN [4], GCNII [5] and GAT [6]) for experiments to answer your question. In addition, we have also conducted experiments based on diverse graph transformer backbones ***in our one-page PDF (Table 1)***. Note that, our observations and conclusions remain the same with the new results. If our paper is accepted, we promise to add these new results to the final version of the paper.
>
> [1] Discovering Invariant Rationales for Graph Neural Networks, ICLR 2022
>
> [2] Causal Attention for Interpretable and Generalizable Graph Classification, KDD 2022
>
> [3] How Powerful are Graph Neural Networks?, ICLR 2019
>
> [4] Semi-Supervised Classification with Graph Convolutional Networks, ICLR 2017
>
> [5] Simple and deep graph convolutional networks, ICML 2020
>
> [6] Graph Attention Networks, ICLR 2018

---

### Official Review · Reviewer_bjaC · 2023-07-11

**Soundness:** 3 good
**Presentation:** 3 good
**Contribution:** 3 good
**Rating:** 4
**Confidence:** 4

**Summary:**

The paper proposes a data augmentation technique on graph datasets. The model consists of two main components: adversarial augmenter and stable feature generator. The adversarial augmenter tries to generate new samples by generating dropping masks for the nodes and edges adversarially, within some augmentation cost. The stable feature generator, on the other hand, tries to identify subsets of graphs that are preserved among all the samples. The authors argue that this construction is more suitable for covariate shift settings compared to the previous graph augmentation methods like DropEdge. Finally, the authors demonstrate the effectiveness of the technique in the experiments.

**Strengths:**

- Data augmentations on graphs is an important area that needs more innovations, as many prominent techniques for data augmentation in other domains, like vision, do not apply to graph data.
- The paper provides explanation details on covariate shift problem in graph setting
- The technique proposed by the authors is interesting. The use of two components: the stable feature generation and adversarial augmenter, on graph data, is new.
- The technique trains the GNN, stable feature generation, and adversarial augmenter together in an alternative way.
- The results demonstrate that the proposed technique outperforms baselines on real-world covariate shift datasets.

**Weaknesses:**

- First, I would like to clarify the authors' claim that "covariate shift is less explored type of OOD" in the introduction. Contrary to this claim, covariate shift (sometimes also called sample selection bias) is a well-studied problem with a tremendous amount of previous and current research works. The efforts to develop algorithms to correct the shift/bias have been made by many previous works. The most notable family of techniques is the important weighting technique. I suggest the author deep dive more into the literature about covariate shift/sample selection bias.
- There are some unclear descriptions of the methods that I would like the author to clarify. Particularly, it's not clear to me how the stable feature generator can achieve the desired results of finding features (or subgraphs) that persist in most graphs in the dataset. I understand its mask construction tries to shelter some of the components in the graph from being perturbed by the augmenter. However, from the loss function, it is not clear how this construction end-up generating mask for common patterns in the dataset, not just creating a unique mask for each graph sample.
- The connection of the paper to covariate shift is also a bit misleading. In the standard covariate shift/sample selection bias setting, there are no need or stable features to exist in the dataset. The covariate shift only requires the sample distribution P(x) between training and testing to be different, while the label conditional distribution P(y|x) remains the same (Shimodaira, 2000; Sugiyama et.al, 2007). The testing sample distribution may or may not have some common characteristics with the training distribution.
- The paper assumes the existence of stable features and actively finds them in the training dataset. In the standard covariate shift setting, even though some of the characteristics commonly occur in the training data, there is no guarantee that they also occur in the test data.
- In short, the authors provide an interesting technique for data augmentation on graph data and demonstrate its effectiveness. However, attributing it to the covariate shift setting is misleading.

Ref:
- H. Shimodaira. Improving predictive inference under covariate shift by weighting the log-likelihood function. Journal of Statistical Planning and Inference, 90(2):227–244, 2000
- Sugiyama, M., Krauledat, M., & Müller, K. R. (2007). Covariate shift adaptation by importance weighted cross validation. Journal of Machine Learning Research, 8(5).


**Questions:**

Please answer my concerns in the previous section.


**Limitations:**

No concern.

---

> ### Author Rebuttal · Authors · 2023-08-09
>
> Thank you for your time and comments! We have responded to your concerns as follows. We sincerely hope that addressing these concerns will help change your rating of our work!
>
> ### Q1.  Covariate shift is a well-studied problem.
>
> - **We're talking about Graph Covariate Shift.** We agree with you that covariate shift is well-studied in general settings [1-4]. But our paper looks at a different and new problem called "graph covariate shift." This is a new area in graph learning and hasn't been studied a lot, as Reviewer ExMB said, "*It is an important OOD area of graph learning*". When we say "covariate shift is a less explored type of OOD," we mean this specific graph learning context. We are very sorry if our words seem to be about the general settings.  And we will thoroughly fix all unclear parts in our paper and make sure it's clear.
> - **Graph OOD is not the same as general OOD.** Traditional OOD problems usually deal with simple tasks like computer vision. For these tasks, inputs are things like variables or image features. But graphs are more complex. This means that graph OOD problems also have to deal with structure distribution shifts, not just shifts in the features. Therefore, it is non-trivial to study covariate shift in graph learning tasks (comments from Reviewer ExMB).
>
> Thanks for your suggestion! If our paper is accepted, we promise to add more discussions about covariate shift, such as [1-4].
>
> ### Q2. The connection to covariate shift is misleading. & The paper assumes the "common patterns" should exist in both training and test distribution.
>
> ### Q3. How the stable feature generator can find stable features in the dataset?
>
> Your concerns (Q2 and Q3) might come from a misunderstanding about "stable features" and the invariance assumption. We first begin with the following two clarifications:
>
> - **Clarification 1: The Invariance Assumption.**  OOD generalization is challenging and even impossible without any assumptions [5]. Hence, the invariance assumption has been introduced by [5] and applied by follow-up works [6-11] as a cornerstone assumption for data generation which enables reasonable problem-solving and analysis for OOD problems. Specifically, stable feature (or invariant feature) causally determines the label and their relationship is invariant across distributions.  Let's use the example of a cow in different backgrounds (grassy or sandy). The cow object is the "stable feature" since the relation from the cow object to the label is invariant across environments (various backgrounds). If we train our model (like an image classifier) on images with a grassy background, it should learn to focus on the cow (stable feature) and ignore the background. This way, it can work well even when we show images with new environments (e.g., sandy background). Similarly, stable features also widely exist in the graph domain (e.g., molecular functional groups [9-11]).
> - **Clarification 2: Stable Features Are Not Always Common Patterns.** The reviewer seems to think that stable features are always common patterns. We didn't make this claim in our paper. This misunderstanding could come from Motif dataset, where the stable feature looks similar a lot. It is a synthetic dataset that are commonly-used in existing studies. They [10, 11] usually use Motif to intuitively check how well a method works. In our work, we only assume the causal mechanisms are invariant [5-11]. We don't say that the stable feature itself always looks the same. For example, in the cow picture, the cow could be different colors or shapes. In molecular graphs (e.g., Molhiv and Molbbbp in our paper), different parts might cause a certain property. Thanks, we will emphasize this in our final paper.
>
> Now let's move on to your two questions:
>
> **Answer to Q2:** In our paper (LINE-110), we provide a definition for covariate shift. Our work aims to address graph covariate shift based on the invariance assumption. This assumption for OOD has been explored in existing studies, such as in Ood-Bench [8] and GOOD [9]. Ood-Bench proves that dealing with covariate shift is challenging and even impossible without any assumptions about data generation. GOOD creates many graph-domain datasets that show graph covariate shift, also based on the invariance assumption. We use GOOD's datasets and therefore follow their settings and assumptions.
>
> **Answer to Q3**: From the above clarifications, we are not looking for "common pattern", but looking for "stable features". Stable features are substructures of the data that can causally determine the label, and their relationship is invariant across environments. The model can make right predictions based on stable features. Now let's check our loss: $\ell(f(T_{\theta_2}(g)), y) + \ell(f(\widetilde{g}), y)$. The first item "$\ell(f(T_{\theta_2}(g)), y)$" requires the model to make correct predictions based on the estimated stable features, and $\ell(f(\widetilde{g}), y)$ requires invariant predictions across different environments. We also confirmed through intuitive visualizations (see Figure 3) and experiments (see Table 3) that we can approximately find stable features.
>
> References:
>
> [1] A Theoretical Analysis on Independence-driven Importance Weighting for Covariate-shift Generalization, ICML 2022
>
> [2] Causally Regularized Learning with Agnostic Data Selection Bias, ACM MM 2018
>
> [3] Rethinking Importance Weighting for Deep Learning under Distribution Shift, NeurIPS 2020
>
> [4] Covariate-Shift Generalization via Random Sample Weighting, AAAI 2023
>
> [5] Invariant models for causal transfer learning, JMLR, 2018.
>
> [6] Invariant risk minimization, ArXiv, 2019.
>
> [7] OOD generalization via risk extrapolation, ICML, 2021.
>
> [8] Quantifying and Understanding Two Dimensions of OOD Generalization, CVPR 2022
>
> [9] GOOD: A Graph OOD Benchmark, NeurIPS 2022
>
> [10] Discovering Invariant Rationales for GNNs, ICLR 2022
>
> [11] Causal Attention for Interpretable and Generalizable Graph Classification, KDD 2022

---

> > ### Comment · Reviewer_bjaC · 2023-08-13
> >
> > Thanks for the authors for the detailed rebuttal.
> >
> > First, I appreciate that the authors will revise the paper to clarify about the focus on graph covariate shift.
> > I do agree that in graph, covariate shift is less studied compared to other OOD types.
> > I still do suggest that the authors provide mode discussion to connect with classical covariate shift setting, e.g., by citing many works on the standard covariate shift areas.
> >
> > I do appreciate the efforts from the authors to clarify my confusions. It partially answers my questions.
> > However, I still have some concerns.
> >
> > I now understand more clearly that the terminology of "stable feature" used by the authors. The stable features causally determines the label and their relationship is invariant across distributions; and they are not necessary always the common patterns.
> >
> > What my concern is that the paper only considers a subset of covariate shift, not the generic covariate shift. Let me illustrate my concern using the example given by the authors:
> >
> > > Let's use the example of a cow in different backgrounds (grassy or sandy). The cow object is the "stable feature" since the relation from the cow object to the label is invariant across environments (various backgrounds). If we train our model (like an image classifier) on images with a grassy background, it should learn to focus on the cow (stable feature) and ignore the background. This way, it can work well even when we show images with new environments (e.g., sandy background).
> >
> > Here, the authors only consider the shifts in the backgrounds, i.e., the test set contains new environments that may not appear in the training set or only represented by a very few samples.
> >
> > Covariate shift in general does not have that restriction (i.e., the restriction that only background features change during deployment). It starts form the concept of "sample selection bias" where there is some underlying bias in the training data vs testing data selection.
> >
> > Let me give a similar example of a covariate shift problem in similar setting.
> >
> > > We have an animal classification problem where the foreground objects are the animal (stable features), and the background may vary. The classifications are multiclass, for example ("cow", "camel", "lion", "jaguar", "llama"). During the training, due to sample selection bias, we can only gather images from animals from African continent and only very few data from outside Africa. However, in the deployment/test setting, the classifiers are tasked to classify animals that mostly come from Latin America. In these settings, the distribution of sample data in training and testing phase vastly differs (e.g., training data will contain more lion and camel samples, where the test data contain more jaguar and llama samples).
> > The relationship between the foreground objects (stable features) and the label in both training and test data does not change. But the distributions of sample itself are drastically different.
> >
> > This generic covariate shift setting is not considered by the model.

---

> > > ### Author Response · Authors · 2023-08-14
> > > **(New rebuttal by authors) Clarification on your misunderstanding of our "cow example".**
> > >
> > > Thank you for taking the time to review our work and for your constructive feedback! We're glad to hear that many of your confusions have been addressed. As graph OOD is a new field, our study is one of the early explorations into the covariate shift. If our paper is accepted, we'll expand on this topic, incorporating **a new Section** in our paper to discuss covariate shift literature more thoroughly. Please see our responses on your remaining concerns:
> > > - **Regarding the cow example.** We apologize for any confusion caused by our cow example. This example is simply a tool to help explain "stable feature" for you. It's not part of our main paper and is just for informal illustration during the rebuttal stage. Our work isn't limited to situations like this example. Instead, we follow the generic covariate shift setting.
> > >
> > > Below we explain the misunderstandings that have arisen between us:
> > > - **Two possible cases in covariate shift.** Our work is based on the generic definition of covariate shift: where $p(x)$ differs between training and testing,  i.e., $p_{train}(x)\neq p_{test}(x)$, but $p(y|x)$ remains constant. The shift in $p(x)$ can arise from:
> > >   - **Case 1**: A shift in environmental features,  $p_{train}(x_{env})\neq p_{test}(x_{env})$. Using the cow example, this is like having different background features in training and test datasets.
> > >   - **Case 2**: A shift in stable features, $p_{train}(x_{sta})\neq p_{test}(x_{sta})$. As in your animal classification example, the stable features themselves can vary between training and test datasets.
> > >
> > > You think we only considered the covariate shift caused by **Case 1**, so it is only a subset of covariate shift. However, we'd like to emphasize that our study isn't limited to this. We follow $p_{train}(x)\neq p_{test}(x)$, which also includes the **Case 2**. Below we give the following evidence.
> > > - **Case 2 in our real-world molecular dataset.** In Molbbbp, we count $p_{train}(x_{sta})$ and $p_{test}(x_{sta})$ in the training and test data, respectively. We use the Chemical Molecular Property Analysis Toolkit (RDKit) to split the functional groups. These functional groups can be regarded as stable features. We performed statistics on their distributions (top 8) in the table below. We can find that **Case 2** obviously exists in our dataset.
> > >
> > > |      |  [OH]  |  [F]   | \[Cl]  | \[Br] | [C(=O)N] | [NH2]  | [n1ccccc1] | \[#6]\[C](=[O])[#6] |
> > > | ----------- | :----: | :----: | :----: | :---: | :------: | :----: | :--------: | :-----------: |
> > > | Training data | 19.66% | 10.20% | 14.77% | 1.41% | 25.41% | 11.07% |  5.48% | 11.99% |
> > > | Test data | 14.77% | 5.57%  | 19.35% | 2.23% | 15.61% | 16.28% | 8.92%  | 6.13% |
> > >
> > > - **Our performance under Case 2.** We design new experiments to support our claim. Our setup focuses on adjusting the selection bias on Motif dataset, primarily based on **Case 2** and **your example**. Specifically, the dataset comprises three classes: "house", "crane", and "cycle". In training dataset, the proportion of "house" and "cycle" is set as $b$, whereas in test dataset, the proportion for them is defined as $1/b$. This setup is very similar to your example: "training data will contain more lion and camel samples, where the test data contain more jaguar and llama samples". We observe that at $b=1$, the distributions $p_{train}(x_{sta})$ and $p_{test}(x_{sta})$ are nearly identical. For $b<1$, these distributions diverge. We adjust different selection biases $b$, and the results are shown in the following table. Our method can also achieve consistent improvements across  $b$. It shows that our work indeed be applied to the generic covariate shift setting, rather than just considering changes in the background features.
> > >
> > > | Method | b=0.7 | b=0.5 | b=0.3 | b=0.1 |
> > > | ------ | :-------: | :-------: | :-------: | :-------: |
> > > | ERM   |   63.88   |   52.34   |   49.56   |   51.59   |
> > > | CAL    |   66.79   |   55.25   |   50.42   |   51.51   |
> > > | GREA  |   64.40   |   59.69   |   54.23   |   55.49   |
> > > | Ours   | **70.43** | **62.33** | **58.79** | **59.98** |
> > >
> > > **We apologize for any confusion stemming from the cow example in our rebuttal.** We will not to include this informal example in the main paper. And we deeply appreciate your constructive feedback, and in response, we've outlined the following revisions for our paper:
> > >
> > > 1. Clarify ambiguous words, notably "covariate shift is a less explored type of OOD".
> > > 2. Introduce a new Chapter/Section dedicated to extensive studies on covariate shift/selection bias.
> > > 3. Provide a precise definition and give descriptions of covariate shift for enhanced clarity.
> > > 4. Incorporate all experimental and statistical insights from our rebuttal.
> > >
> > > If our paper is accepted, we promise that we will thoroughly improve the paper according to the above to-do list. Finally, thank you again for your reviews and precious time! We sincerely hope this reply can solve your remaining concerns and change your negative rating on our work!

---

> > > > ### Author Response · Authors · 2023-08-17
> > > > **(New response by authors) Looking forward to your reply!**
> > > >
> > > > Thank you for your valuable comments on our submission! According to your suggestions, we will
> > > > - **clarify ambiguous words,**
> > > > -  **introduce a new Section for classical covariate shift/sample selection bias,**
> > > > -  **provide a precise definition or clear description of our covariate shift setting,**
> > > > -  **incorporate all experimental and statistical results from our rebuttal.**
> > > >
> > > > These insightful suggestions help us to substantially improve the coherence and significance of our submission! We hope that these improvements will be taken into consideration.
> > > >
> > > > If our response has resolved your concerns on our paper, we will greatly appreciate it if you could re-evaluate our paper. Should you have any further questions or need additional clarification, please know that we are eager and prepared to continue our discussions.

---

> > > > > ### Comment · Reviewer_bjaC · 2023-08-18
> > > > >
> > > > > Thanks for the authors for the explanations on the cases of covariate shifts, and providing additional experiments on the second case of covariate shift.
> > > > >
> > > > > However, my perception that the paper focus only on Case 1 did not only come from the cow examples, but mostly directly from the writing in the paper.
> > > > > There are many statements in the paper that indicate the focus of the paper is only on Case 1. For example:
> > > > >
> > > > > - Line 8
> > > > > > covariate shift, implying the presence of new environmental features in test data. However, most strategies, such as invariant learning or graph augmentation, typically struggle with limited training environments or perturbed stable features, thus exposing limitations in handling the covariate shift issue
> > > > >
> > > > > - Line 14
> > > > > > Specifically, given the training data, AIA aims to extrapolate and generate new environments, while concurrently preserving the original stable features during the augmentation process.
> > > > >
> > > > > - Line 37
> > > > > > covariate shift characterizes that the environmental features in test data are unseen in training data
> > > > >
> > > > > - Line 45
> > > > > > Whereas, scaffolds (e.g., carbon rings) are usually patterns irrelevant to the molecule properties, which can be seen as environmental features [1, 14]. In practice, we often need to use molecules collected in the past to train models, hoping that the models can predict the properties of molecules with new scaffolds in the future
> > > > >
> > > > > - Line 69
> > > > > > Hence, we naturally ask a question: “Compared to the training data, can we generate new data that satisfy two conditions: 1) having new environments; 2) keeping the original stable features unchanged?”
> > > > >
> > > > > - Line 72
> > > > > > Towards this end, we introduce two intuitive principles for graph augmentation: environmental feature discrepancy and stable feature consistency. The discrepancy principle promotes the exploration  of new environments beyond the scope of training data, while the consistency principle seeks to maintain the integrity of stable features during augmentation.
> > > > >
> > > > > - Line 81
> > > > > > As depicted in Figure 1, AIA primarily augments environmental features  while leaving the stable elements unchanged. Our approach equips the graph classifier model with an enhanced ability to identify stable features in new environments and effectively mitigate the covariate shift issue
> > > > >
> > > > > - Supplementary, Line 14
> > > > > > Covariate shift Ptr(G) ≠ Pte(G), Ptr(Y ∣G) = Pte(Y ∣G). If there exist environmental features in the test distribution that the model has not seen during training, it will also result in a large performance drop. This unseen distribution shift is well known as covariate shift [7] or diversity shift [5]. It means that the environmental features in test data are unseen in training data, which leads to Ptr(G) ≠ Pte(G). In Definition 2.1, we also quantitatively measure the covariate shift between Ptr(G) and Pte(G).
> > > > >
> > > > > And in many other places.

---

> > > > > > ### Comment · Reviewer_bjaC · 2023-08-18
> > > > > >
> > > > > > I would also say, that the design of the proposed algorithm largely motivated by the Case 1 only, i.e. shift in the environmental features.
> > > > > > This is reflected on the design of the Adversarial Augmenter that adversarially augment the data, while using Stable Feature Generator to keeps stable feature consistency. This will make the model to be able to adapt to the shift in the environmental features.
> > > > > >
> > > > > > I also do acknowledge that the concept in Section 2, i.e. Definition 2.1 and Problem 2.2 apply to both Case 1 and Case 2. Only after Section 3, we can see the paper starts to focus on Case 1, by the introduction of Principle 3.1 and Principle 3.2.

---

> > > > > > > ### Author Response · Authors · 2023-08-19
> > > > > > > **Response to Reviewer bjaC: Contextualizing Our Work in the Graph Domain**
> > > > > > >
> > > > > > > We greatly appreciate the Reviewer bjaC's patience and time to respond to us. We have noted that most of the examples or settings you referred to are in the Computer Vision (CV) field. We appreciate your thoughts but respectfully emphasize that **our research is specifically focused on the Graph Domain**, where the setting we consider is highly common in practice.
> > > > > > >
> > > > > > > Below, we outline the following points to support our claim:
> > > > > > >
> > > > > > > 1. **Distinct Causes of Covariate Shift in Graph Domain**. The main cause of the covariate shift in the graph domain is the presence of "new environmental features." For instance, in molecular datasets, the molecular scaffold serves as environmental features. Often, we are tasked with predicting the properties of molecules that possess different scaffolds. And this OOD setting is frequently and consistently indicated in the official Open Graph Benchmark (OGB) dataset [1]. For further details, please refer to the OGB [1] or MoleculeNet [2].
> > > > > > >
> > > > > > > 2. **Alignment with Standard Setting from OOD Benchmarks in Graph Domain**. GOOD [3] is a widely acknowledged OOD benchmark specific to the graph domain. This work also confirms that our setting aligns with the most common scenarios encountered in graph domains. It states that covariate shift predominantly arises from differences in environmental features (as per their paper: "input features that are not associated with Y"), subsequently establishing different domains based on these variances in environmental features, thereby creating covariate shift. Our work utilizes their datasets and adheres rigorously to their prescribed settings.
> > > > > > >
> > > > > > > 3. **Understanding of Our Research in Line with Peer Graph Domain Studies.** We would like to highlight that our work is in harmony with the perspectives presented in related research within the graph domain. For instance,
> > > > > > >
> > > > > > >    - DIR [4] mentions that "..different distributions elicit different environments...";
> > > > > > >    - GREA [5] states "environment subgraphs can be viewed as natural noises...";
> > > > > > >    - CAL [6] notes "these (environment) features might easily change outside the training distribution".
> > > > > > >
> > > > > > >    These sentences from respected works in the field illustrate that "the shift in environmental features" is a widely recognized phenomenon in the graph field. Consequently, current methods, akin to our own, are increasingly being designed with consideration for environmental features, through strategies such as data augmentation or intervention.
> > > > > > >
> > > > > > > 4. **Quantitative Evidence from Our Graph Dataset**. We quantitatively measure graph covariate shift (GCS) of the environmental and stable features in our dataset. The results clearly indicate that the main cause of the observed covariate shift is attributed to variations in environmental features.
> > > > > > >
> > > > > > > | Graph Covariate Shift (GCS) | Motif | Molbbbp |
> > > > > > > | --------------------------- | :---: | :-----: |
> > > > > > > | Full features               | 0.557 |  0.419  |
> > > > > > > | Stable features             | 0.022 |  0.130  |
> > > > > > > | Env features                | 0.539 |  0.407  |
> > > > > > >
> > > > > > > Finally, we would like to emphasize that the main contribution of our work lies in **designing a novel algorithm that is specially tailored for the Graph Domain**, rather than considering all possible general situations such as in CV data, which is beyond the scope of our paper (graph domain). Therefore, we need to consider the actual situation in the graph field to design a reasonable algorithm for generalization. Nevertheless, we appreciate the situation you mentioned, and we will include a detailed discussion in our final version.
> > > > > > >
> > > > > > > We respectfully request that these points be considered during your evaluation of our work. Thank you once again for your invaluable time and insightful feedback!
> > > > > > >
> > > > > > > [1] Open Graph Benchmark: Datasets for Machine Learning on Graphs, NeurIPS 2020
> > > > > > >
> > > > > > > [2] MoleculeNet: A Benchmark for Molecular Machine Learning, arXiv preprint, arXiv: 1703.00564, 2017
> > > > > > >
> > > > > > > [3] GOOD: A Graph Out-of-Distribution Benchmark, NeurIPS 2022
> > > > > > >
> > > > > > > [4] DISCOVERING INVARIANT RATIONALES FOR GRAPH NEURAL NETWORKS, ICLR 2022
> > > > > > >
> > > > > > > [5] Graph Rationalization with Environment-based Augmentations, KDD 2022
> > > > > > >
> > > > > > > [6] Causal Attention for Interpretable and Generalizable Graph Classification, KDD 2022

---

> > > > > > > > ### Comment · Reviewer_bjaC · 2023-08-20
> > > > > > > >
> > > > > > > > Dear authors. Thank you for the explanations.
> > > > > > > >
> > > > > > > > The definition of covariate shift only states that $P_{test}(X) \neq P_{train}(X)$, while the predictive conditional distribution remains the same $P_{test}(Y|X) = P_{train}(Y|X)$. No additional restriction on the shift on P(X).
> > > > > > > > This applies to **any** domain area, not just computer vision or tabular data, but also graph domain as well.
> > > > > > > >
> > > > > > > > The covariate shift, or sample selection bias problem, is usually studied in the literature using artificially created datasets that incorporate this bias by selecting a deployment domain that makes the data distributions between training and test differ, for example, the deployment location of the model (Africa vs. south America, in my previous example).
> > > > > > > >
> > > > > > > > This construction is also used in the benchmark datasets of the GOOD (NeurIPS 2022) paper. Figure 1 from GOOD papers illustrates many examples of covariate shifts in graphs, where the training and test examples are from different domains. For example, the ArXiv paper dataset, that predict the subject area of the paper, with time (publication year) as the domain selected for introducing covariate shift. The time variable should not influence the prediction, but it may create different data distributions of each subject area; based on the selection of certain time of publications. The distributions of subject areas of papers (in CS) from the year before 2012 and after 2018 may differ a lot, for example, due to the rising popularity of deep learning.

---

> > > > > > > > > ### Author Response · Authors · 2023-08-20
> > > > > > > > > **Response to Reviewer bjaC: We Fully Get Your Points**
> > > > > > > > >
> > > > > > > > > Dear Reviewer bjaC, we really appreciate your patience and time in replying to us these days!
> > > > > > > > >
> > > > > > > > > First of all, we fully agree with you on the following points:
> > > > > > > > >
> > > > > > > > > - We agree that general covariate shift should not consider environment or stable feature shifts.
> > > > > > > > > - We agree  $P_{train}(X) \neq P_{test}(X), P_{train}(Y|X) = P_{test}(Y|X)$ in covariate shift.
> > > > > > > > > - We agree that it applies to **any** domain area, from computer vision and tabular data to the graph domain.
> > > > > > > > >
> > > > > > > > > Now our disagreement converges to "the design of our method is largely motivated by environment shift". We agree that our design is indeed motivated by the special case of covariate shift caused by environment shift. Your suggestion is correct, and we have modified our claim in the revision. However, we would like to clarify the following points:
> > > > > > > > >
> > > > > > > > > 1. **Real-world Scenarios in Graph Classification.** Our work is deeply rooted in real-world scenarios observed in **graph classification tasks**. Specifically, in numerous instances in graph classification, covariate shifts, as seen in various research works [1-6], predominantly originate from the environment shifts. As illustrated earlier, in molecular graphs, scaffolds are frequently interpreted as environmental features, with the majority of shifts emerging from scaffolds. Numerous efforts on graph classification based on these datasets (OGB [1], MoleculeNet [2], GOOD [3]) further agree with this point.
> > > > > > > > >
> > > > > > > > > 2. **General settings can be used for graph domains, but we believe our settings are more realistic.** We agree that the case you mentioned can apply to the graph domain. As you said "***artificially created datasets that incorporate this bias***", so we can also "artificially" create the shift you pointed out in the graph classification dataset. But in practice, we believe that the situation we consider may be more realistic.
> > > > > > > > >
> > > > > > > > > 3. **GOOD creates graph classification datasets mainly based on irrelevant features.** Please refer to [Section 3.1 covariate shift] in GOOD [3], it says
> > > > > > > > >
> > > > > > > > >    > covariate shift can only happen on input features that are not associated with Y. Therefore, with prior knowledge, we can manually select and shift one or several of these irrelevant features, $X_{ind}$, to build covariate splits. Different $X_{ind}$ feature values indicate different domains, and each domain can be viewed as a split. For instance, in the graph ColoredMNIST dataset in which we distinguish hand-written digits with colors, the color is irrelevant with labels. Thus, in our covariate splits, digits with different colors belong to corresponding color domains, and each domain becomes a split.
> > > > > > > > >
> > > > > > > > > Therefore, the covariate shift in the GOOD graph classification dataset we used is mainly due to irrelevant features.
> > > > > > > > >
> > > > > > > > > Finally, we acknowledge that our work is indeed motivated by the special case of covariate shift caused by environment shift. Your suggestion is correct, and based on your feedback, we've updated our claims in the revised paper. While we agree that the general settings you've pointed out are crucial and plan to explore them in future research, we also believe our research on our case is still significant and meaningful in graph classification domain.
> > > > > > > > >
> > > > > > > > > We genuinely hope you'll consider these points during your re-evaluation of our work. We're fortunate to have such a thoughtful reviewer! Once again, thank you for your valuable insights and time!
> > > > > > > > >
> > > > > > > > > [1] Open Graph Benchmark: Datasets for Machine Learning on Graphs, NeurIPS 2020
> > > > > > > > >
> > > > > > > > > [2] MoleculeNet: A Benchmark for Molecular Machine Learning, arXiv preprint, arXiv: 1703.00564, 2017
> > > > > > > > >
> > > > > > > > > [3] GOOD: A Graph Out-of-Distribution Benchmark, NeurIPS 2022
> > > > > > > > >
> > > > > > > > > [4] DISCOVERING INVARIANT RATIONALES FOR GRAPH NEURAL NETWORKS, ICLR 2022
> > > > > > > > >
> > > > > > > > > [5] Graph Rationalization with Environment-based Augmentations, KDD 2022
> > > > > > > > >
> > > > > > > > > [6] Causal Attention for Interpretable and Generalizable Graph Classification, KDD 2022

---

> > > > > > > > > > ### Comment · Reviewer_bjaC · 2023-08-21
> > > > > > > > > >
> > > > > > > > > > Dear Authors.
> > > > > > > > > > Thank you for acknowledging the concerns that I have.
> > > > > > > > > >
> > > > > > > > > > Regarding to your statement:
> > > > > > > > > > > Considering that in real-world graph classification tasks, covariate shift mostly comes from changes in environmental features.
> > > > > > > > > >
> > > > > > > > > > I disagree with it. As mentioned in GOOD datasets, many covariate shift problems in graph are not just based on environmental features.
> > > > > > > > > > I could also imagine that there are many graph applications out there that their training and testing have different samples distributions, such as in e-commerce or fraud detections.
> > > > > > > > > >
> > > > > > > > > > Going back to my original comment,
> > > > > > > > > > > In short, the authors provide an interesting technique for data augmentation on graph data and demonstrate its effectiveness. However, attributing it to the covariate shift setting is misleading.
> > > > > > > > > >
> > > > > > > > > > I still stick with this assessment.
> > > > > > > > > >
> > > > > > > > > > As the method works of a subset of covariate shift problem, it is not wise to 'market' it as a method to handle general covariate shift. Perhaps changing the paper title and how the proposed method is 'marketed' to, for example, a method to handle environmental shift, will resolve many of the confusions.

---

> > > > > > > > > > > ### Author Response · Authors · 2023-08-21
> > > > > > > > > > >
> > > > > > > > > > > Dear Reviewer bjaC, thanks for your reply!
> > > > > > > > > > >
> > > > > > > > > > > OOD/distribution shift on graph is an emerging research field. How to define and divide specific shifts is still an open problem. As an initial work in graph covariate shift, we follow the universally acknowledged and widely accepted consensus in this field, that distribution shift may mainly arise from environmental features, to design our work. We agree with you that in addition to environmental feature shifts, there may indeed be covariate shifts caused by other features. We really appreciate your comments, which help to expand the scope of our work!
> > > > > > > > > > >
> > > > > > > > > > > In addition, we would like to clarify that our task is graph classification/graph prediction. The division of the graph classification dataset in the GOOD dataset is indeed mainly to define different domains according to the different environmental features. For example in the original paper they say:
> > > > > > > > > > >
> > > > > > > > > > > - GOOD-Motif: "we generate graphs using five label irrelevant base graphs (wheel, tree, ladder, star, and path)"; "To create covariate splits, we select the base graph type and the size as domain features."
> > > > > > > > > > > - GOOD-CMNIST: "in covariate shift split, we color digits with 7 different colors", "we color digits according to their domains".
> > > > > > > > > > > - GOOD-HIV: "We design splits based on two domain selections, namely, scaffold and size."
> > > > > > > > > > >
> > > > > > > > > > > Regarding what you said, e-commerce or fraud detections in the graph classification task with general covariate shift, we respectfully request if you can provide some related references, and we will put them in our paper to discuss potential extension.
> > > > > > > > > > >
> > > > > > > > > > > Finally, we agree with you that we do not attribute our approach to general covariate shift settings. In response to your valuable feedback, we have revised our paper to underscore that our primary emphasis lies on addressing the covariate shift mainly induced by environmental features.
> > > > > > > > > > >
> > > > > > > > > > > Thanks again for your valuable advice!

---

> > > > > > > > > > > > ### Author Response · Authors · 2023-08-21
> > > > > > > > > > > > **Supplementary Reply by Authors**
> > > > > > > > > > > >
> > > > > > > > > > > > Considering that the Author-Reviewer discussion stage is about to end, we may not have the opportunity to conduct further rounds of discussions with you, so we add another supplementary reply. We sincerely hope that you can understand our situation.
> > > > > > > > > > > >
> > > > > > > > > > > > Graph OOD is still a relatively new and immature field. Due to the complexity of graph data (node features, edge features, and structural information that may exist in the data), the specific types, specific definitions, and causes of distribution shifts on the graph are still a relatively open topic. And it is also becoming a hot topic in the GNN community recently.
> > > > > > > > > > > >
> > > > > > > > > > > > I understand that you are confused that we analyze the distribution shift from the granularity of features, such as considering environmental features or stable features. The reason for our analysis is that in the current graph OOD field, especially in the graph classification task, such analysis is a generally accepted consensus. For example, in the baseline studies referenced by our paper:
> > > > > > > > > > > >
> > > > > > > > > > > > - The causal part and non-causal part are defined in DIR [1];
> > > > > > > > > > > > - Rationale and environment subgraph are defined in GREA [2];
> > > > > > > > > > > > - The causal feature and shortcut feature are defined in CAL [3];
> > > > > > > > > > > > - Invariant subgraph and variant subgraph are defined in GIL [4];
> > > > > > > > > > > > - The causal subgraph and bias subgraph are defined in DisC [5];
> > > > > > > > > > > > - Invariant part and variant part are defined in CIGA [6];
> > > > > > > > > > > > - The label-relevant subgraph and label-irrelevant subgraph are defined in GSAT [7].
> > > > > > > > > > > >
> > > > > > > > > > > > We speculate that the reason for this consideration is that these methods are inspired by invariant learning [9,10], which facilitates a clearer and concise understanding of the causes of distribution shifts in graphs.
> > > > > > > > > > > >
> > > > > > > > > > > > Another reason may be that such an analysis process is often also referred to as "intrinsic interpretability" in some graph OOD studies [1,2,3,7]. Analyzing the problems at the feature level is more conducive to understanding the internal working mechanism of the GNN, such as finding "subgraphs" in the field of graph explainability [8], which can further guide researchers to design more generalization and robustness GNNs.
> > > > > > > > > > > >
> > > > > > > > > > > > Considering that we are an initial work on graph covariate shift (exploring new environments), we design our model following these consensus of the current mainstream graph OOD field. After discussing with you these days, we agree that the case we consider is indeed an environmental covariate shift.
> > > > > > > > > > > >
> > > > > > > > > > > > We are very sorry for any confusion caused by our setting. We promise to emphasize this point in our paper. We can also modify our title according to your suggestions, for example, modify it to "Unleashing the Power of Graph Data Augmentation on Environmental Covariate Shift". Then we can add a definition of "Environmental Covariate Shift" to our main paper. We believe this is also a non-trivial contribution to the graph OOD community. In addition, we will also conduct in-depth research on the general covariate shift on the graph in future work, such as trying to explore how to transfer technologies such as "important weighting" in the CV field to further expand the scope of application of our method.
> > > > > > > > > > > >
> > > > > > > > > > > > Finally, we sincerely hope these replies can earn you some appreciation for our work. And we truly appreciate your understanding and patience!
> > > > > > > > > > > >
> > > > > > > > > > > > [1] Discovering Invariant Rationales for GNNs, ICLR 2022
> > > > > > > > > > > >
> > > > > > > > > > > > [2] Graph Rationalization with Environment-based Augmentations, KDD 2022
> > > > > > > > > > > >
> > > > > > > > > > > > [3] Causal Attention for Interpretable and Generalizable Graph Classification, KDD 2022
> > > > > > > > > > > >
> > > > > > > > > > > > [4] Learning Invariant Graph Representations for Out-of-Distribution Generalization, NeurIPS 2022
> > > > > > > > > > > >
> > > > > > > > > > > > [5] Debiasing Graph Neural Networks via Learning Disentangled Causal Substructure, NeurIPS 2022
> > > > > > > > > > > >
> > > > > > > > > > > > [6] Learning Causally Invariant Representations for Out-of-Distribution Generalization on Graphs, NeurIPS 2022
> > > > > > > > > > > >
> > > > > > > > > > > > [7] Interpretable and Generalizable Graph Learning via Stochastic Attention Mechanism, ICML 2022
> > > > > > > > > > > >
> > > > > > > > > > > > [8] Parameterized Explainer for Graph Neural Network, NeurIPS 2020
> > > > > > > > > > > >
> > > > > > > > > > > > [9] Invariant models for causal transfer learning, JMLR, 2018.
> > > > > > > > > > > >
> > > > > > > > > > > > [10] Invariant risk minimization, ArXiv, 2019.

---

### Official Review · Reviewer_reTY · 2023-07-27

**Soundness:** 3 good
**Presentation:** 3 good
**Contribution:** 3 good
**Rating:** 7
**Confidence:** 4

**Summary:**

This paper proposes to enable GNNs handle covariate shifts through the use of synthetic augmentations. An inherent challenge in construction task-specific augmentations is to appropriately handle style (or environment features) and content (or stable features). While other existing works have also emphasized the need to promote task-relevant invariances (e.g., Analyzing data-centric properties for graph contrastive learning, Neurips 2022), this paper focuses on automatically identifying those features through an adversarial training strategy. Results on graph classification benchmarks demonstrate the benefits of AIA.

**Post Author Rebuttal**:The authors have reasonably responded to all my concerns. Hence, I raise my score!

**Strengths:**

+ The paper is well written and easy to follow. The problem is clearly laid out and the I personally liked the organization of the experiments in the form of research questions.
+ Though the proposed algorithm builds upon existing formalisms and ideas from the certified robustness literature, I believe the extension and its implementation to this problem is sufficiently novel.
+ The theoretical analysis is intuitive and well presented.
+ Experiment results and the hyperparameter study provide a convincing demonstration of the proposed approach.

**Weaknesses:**

1. At the outset, the idea of splitting the adversarial augmenter and stable feature generator with independent (learnable) masks appears challenging to solve. There is a risk that the same entries can be picked as relevant in both the masks. While the regularizer checks for the ratio, will it benefit to include an explicit constraint to make the masks disjoint? This is often done in state-of-the-art source separation algorithms that aim to split the different sources from a given observation.
2. It will be beneficial if the authors can intuitively explain how this approach is able to handle size generalization.
3. More insights into the results will help. For example on the Motif benchmark, when compared to the baselines, the bigger benefits are observed in the base setting. Why is that?
4. While GIN has been used for all experiments, will the benefits persist with a graph transformer?

**Questions:**

see weaknesses

---

> ### Author Rebuttal · Authors · 2023-08-09
>
> Thank you for rating our paper as "well written and easy to follow", "clearly", "sufficiently novel", and "well presented"! According to your concerns, we provide the following responses.
>
> ### Q1: Concerns about adversarial augmenter, stable feature generator, masks and regularizer.
>
> - **Difference between stable and adversarial mask.** We utilize two distinct modules to create stable and adversarial masks. The stable mask aims to highlight the stable features in the graph data. Ideally, we'd like the mask value for the stable feature to be 1, while the other parts are 0. The adversarial mask, on the other hand, is used to modify the data. With this mask, we aim to disrupt the graph features, such as by removing certain nodes or edges, in order to create out-of-distribution data.
>
> - **Preventing mask overlap or disjoint.** We understand your concern about the potential for conflict between the two masks. To address this, we use a strategy to combine the masks in a way that the adversarial perturbation doesn't harm the stable features. This is explained in lines 207-218 of the original paper. Basically, the stable mask ($M_{sta}$) highlights the stable regions. Anything not covered by this mask, represented as $1-M_{sta}$, is the complementary part. The adversarial mask ($M_{adv}$) represents the adversarial perturbation. By applying the operation $(1-M_{sta})\odot M_{adv} + M_{sta}$, we apply the adversarial perturbation to the complementary parts "$(1-M_{sta})\odot M_{adv}$", while preserving the stable features "$+ M_{sta}$" . This ensures that the final augmented data includes both new environmental features and original stable features.
>
> - **Purpose of regularization term.** The regularization term is used to guide the masks to closely match our expectations (i.e., the mask value should be near 0 or 1). This prevents the masks from converging to a trivial solution during training, thus helping to avoid overlap or conflict between the masks. For the stable mask, we aim for it to highlight the stable feature, and for its value to be close to 0 or 1. We set the stable ratio as $\lambda_s$ and design two regularization terms to enforce this. For the adversarial mask, we limit its ratio to 1 to prevent it from creating excessive perturbations, like deleting all components to optimize the adversarial goal.
>
> ### Q2. How does this approach handle size generalization?
>
> Great question! Current studies, especially invariant learning, suggest that size distribution shift occurs due to (size) differences in environmental features during training and testing. Let's take molecular graphs as an example. The size of scaffolds, such as carbon chains or rings, can greatly vary. Our method aims to maintain stable features while modifying environmental features to create new graphs. During training, the adversarial mask might alter the graph size by removing varying numbers of nodes or edges (as seen in Figure 2 or Figure 3). This way, the model can handle graphs with different environmental feature sizes, reducing sensitivity to graph size. Therefore, the model can generalize to unseen graphs of different sizes during testing. We appreciate your suggestion and will include this explanation in the final version for clarity.
>
> ### Q3. More insights into the results. For example, the bigger benefits are observed in the Motif (base) setting.
>
> You've made a great observation. Our method shows significant improvements on Motif (base) mainly because it experiences larger covariate shift. We've quantified this covariate shift (GCS) for different datasets (as shown in Table 2). The GCS of Motif (base) is 0.557, which is larger than others. As our method is specifically designed to tackle covariate shift, and other baseline methods have limitations in this regard, our improvement is more noticeable on Motif (base). Further, upon reviewing our results in light of your suggestion, we've added two additional insights:
>
> - Our method outperforms GREA and attains the best performance in terms of size generalization, even though GREA, another data augmentation method, also performs well (second place on motif and molhiv). GREA uses two types of coarse-grained augmentation, namely, environment replacement and removal, which might produce new graphs of varying sizes. In contrast, our method applies a more detailed environmental feature augmentation, specifically by removing certain nodes or edges. Therefore, it considers a broader size scope than GREA, which further improves the performance.
> - Invariant learning methods like DIR, CAL, and DisC struggle to perform well as they find it challenging to generate new environmental features. However, covariate shift is very evident in these datasets (see Table 2). By creating new environments during data augmentation, our method ensures better generalization under covariate shift.
>
> Thank you for your suggestions. We will include these insights in the final version of our paper.
>
> ### Q4. Will the benefits persist with a graph transformer?
>
> Indeed, they do. We conduct experiments on Molbbbp using four different graph transformers [1-4] as shown **in our one-page PDF** (Table 1). The results show that our method continues to provide benefits across these diverse graph transformers.
>
> [1] Do Transformers Really Perform Bad for Graph Representation? NeurIPS 2021
>
> [2] Representing Long-Range Context for Graph Neural Networks with Global Attention, NeurIPS 2021
>
> [3] Rethinking Graph Transformers with Spectral Attention, NeurIPS 2021
>
> [4] Recipe for a General, Powerful, Scalable Graph Transformer, NeurIPS 2022

---

### Author Rebuttal · Authors · 2023-08-09

# Global Response and One-page PDF from Authors

We appreciate all the reviewers' efforts for reviewing this submission. We are delighted that our paper was noted for being "*well written and easy to follow*", "*clear",* "*sufficiently novel*" by **Reviewer reTY**; "*interesting*", "*new*" by **Reviewer bjaC**; offering "*new insights*", "*novel with good intuitions*" by **Reviewer ExMB**; and "*well-motivated*" by **Reviewer J9ZS**. To address the reviewers' concerns, we have included new experimental results and a method comparison table in our ***one-page PDF*** below. Here, we summarize the main points raised by the reviewers and our responses.

- **[@Reviewer bjaC], Covariate shift & Invariance Assumption.** We understand that your main concern stems from a misunderstanding of our proposed "stable feature". We have provided a thorough explanation for you and clarified the relationship between our work and the invariance assumption.  We apologize for any confusion caused by vague phrases such as "covariate shift is a less explored type of OOD", and will thoroughly revise our paper. Finally, we sincerely hope to engage in further discussions to address your concerns and improve your rating of our work.

- **[@Reviewer J9ZS], Novelty & Motivation.** We understand that you question the novelty and motivation of our graph augmentation method. However, other reviewers have strongly recognized the novelty of our approach. Transferring general augmentation methods to graph data is challenging due to its non-Euclidean nature. We have also compared our approach with other similar methods in the graph domain (***Table 3 in PDF***). For your interest in our augmentation diversity experiments, we have included the results in ***Table 4 in PDF***. If our paper is accepted, we'd be glad to include all the running time and model size results (***Table 5 in PDF***) in our appendix. We sincerely hope that addressing these concerns will help change your rating of our work.


- **[@Reviewer J9ZS, ExMB], Complexity & Running Time & Model Size.** Thanks to the modest size of the two extra networks we implemented and our one-step adversarial training, our approach maintains manageable complexity and model size parameters. For the running time and model size comparisons (***Table 5 in PDF***), it is evident that our method achieves a preferable performance-complexity balance compared to other baselines.

- **[@Reviewer reTY, ExMB], Different Backbones.** We conducted experiments using a broader range of backbone models (***Table 1 and 2 in PDF***), including GIN, GCN, GCNII, GAT, and various Graph transformers. The results confirm that our findings and conclusions remain consistent across different backbones.

---

> ### Author Response · Authors · 2023-08-21
> **New Global Response by Authors**
>
> # New Global Response by Authors
>
> First of all, thank all reviewers for their time and efforts on our submission! Considering that the Author-Reviewer discussion period is coming to an end, we send this message to summarize the discussion results and improvements to our paper.
>
> 1. **The current status of the Author-Reviewer discussion.** We thank all reviewers for their positive ratings of our work. Below we summarize our discussions and feedbacks with the reviewers.
>    - **Reviewer reTY and Reviewer J9ZS**:  We believe our rebuttal has addressed your concerns, considering that you have further improved your ratings of our work. Thank you very much for your great support!
>    - **Reviewer ExMB**: We have provided you with detailed experimental results about backbones and an explanation of the training efficiency. Considering your positive rating, we believe we have addressed your concerns. Thank you so much for your positive rating!
>    - **Reviewer bjaC**: We're glad that we have addressed most of your concerns, and eventually your concerns converged on a small issue: "Our work is motivated by environmental feature shifts". To address this point, we have provided detailed explanations below in your comment box. OOD on graph is an emerging research field. How to define and divide specific shifts on graphs is still an open problem. As an initial work in graph covariate shift, we follow the general consensus in this field to design our work. We agree with you that we do not attribute our approach to general covariate shift settings. And we have revised our paper to underscore that our primary emphasis lies on addressing the environmental covariate shift. Thank you so much for your constructive and valuable comments!
>
> 2. **Improvements to our submission.** Below we summarize our improvements to our paper based on the suggestions of all reviewers.
>    - Clarify ambiguous words;
>    - Introduce a new Section for classical covariate shift/sample selection bias;
>    - Underscore that our work lies on covariate shift from environmental features (e.g. title/definition);
>    - Add more results and comparisons from our one-page PDF and rebuttal;
>    - Add more discussions on our complexity, runtime, and model size;
>    - Add more insights for our experimental observations;
>
> If our paper is accepted, we will revise our paper based on the to-do list above.
>
> Finally, we once again thank the reviewers for their efforts to improve the quality of our paper!
>
> We are also very grateful to the reviewers for their strong support for our submission!

---

### Comment · Area_Chair_Ra7X · 2023-08-17
**AC discussions**

The reviews and discussions of this paper are mixed.

Reviewers: please read author rebuttals and discuss.

Reviewer bjaC: could you let us know if your concerns have been addressed?

AC

---

### Decision · Program_Chairs · 2023-09-21

**Decision:**

Accept (poster)

**Comment:**

This paper received four reviews, of which three are quite positive. There have been a lot of discussions during rebuttal and major concerns have been addressed.